# Endogenous noise of neocortical neurons correlates with atypical sensory response variability in the *Fmr1*$^{-/y}$ mouse model of autism

Arjun A. Bhaskaran [1,2,3,4], Théo Gauvrit [1,2,4], Yukti Vyas[1,2], Guillaume Bony[1,2], Melanie Ginger[1,2] & Andreas Frick [1,2] ✉

Excessive neural variability of sensory responses is a hallmark of atypical sensory processing in autistic individuals with cascading effects on other core autism symptoms but unknown neurobiological substrate. Here, by recording neocortical single neuron activity in a well-established mouse model of Fragile X syndrome and autism, we characterized atypical sensory processing and probed the role of endogenous noise sources in exaggerated response variability in males. The analysis of sensory stimulus evoked activity and spontaneous dynamics, as well as neuronal features, reveals a complex cellular and network phenotype. Neocortical sensory information processing is more variable and temporally imprecise. Increased trial-by-trial and inter-neuronal response variability is strongly related to key endogenous noise features, and may give rise to behavioural sensory responsiveness variability in autism. We provide a novel preclinical framework for understanding the sources of endogenous noise and its contribution to core autism symptoms, and for testing the functional consequences for mechanism-based manipulation of noise.

Accurate neural processing of sensory information is fundamental for human perception, higher cognitive abilities, and interaction with our environment. Individuals with neurodevelopmental disorders such as Fragile X syndrome (FXS) and autism spectrum disorder (ASD) commonly report differences in their perception of sensory information. In the case of autism, altered sensory perception has cascading effects on, and is predictive of, other core symptoms[1–3]. Consequently, sensory symptoms are now included as a core diagnostic criterion for ASD in the Diagnostic and Statistical Manual of Mental Disorders[4]. However, there is a paucity of preclinical studies investigating the neurobiological underpinnings of atypical sensory response variability in autism[5].

Excessive inter-individual and trial-by-trial variability of neural responses are hallmarks of noisy sensory processing in autistic individuals[6–12]. An emerging model suggests that exaggerated variability of sensory-evoked responses is the result of a "noisy" brain state, rendering sensory information processing less reliable, as shown by clinical studies[6,7,13–16]. Consequently, autistic individuals exhibit marked heterogeneity in their perception and responsiveness to sensory input[3,17] and temporal sensory processing issues[18,19]. Altered sensory perception also affects situational predictions that are based on prior sensory experience[8].

The concept of "noise", however, is poorly defined for the field of clinical neuroscience. This makes describing its underlying

[1]INSERM, U1215 Neurocentre Magendie, 33077 Bordeaux, France. [2]University of Bordeaux, 33000 Bordeaux, France. [3]Present address: Department of Psychiatry, Djavad Mowafaghian Centre for Brain Health, University of British Columbia, Vancouver, BC, Canada. [4]These authors contributed equally: Arjun A. Bhaskaran, Théo Gauvrit. ✉e-mail: andreas.frick@inserm.fr

mechanisms and their relationship with atypical sensory information processing challenging[7,9,15,16,20–23]. In addition, tests of the sources and constituents of endogenous neural noise and its consequences for sensory information processing remain incomplete in human experiments due to the limited resolution of non-invasive, large-scale physiological measures[23,24].

Here, we consider the hypothesis that internally generated or endogenous neural noise drives atypical sensory response variability. We use the term endogenous noise to describe core neural parameters that emerge from mechanism-derived alterations in cellular or network properties. To overcome the aforementioned limitations and to test our prediction, we turned to the *Fmr1*[−/y] mouse, an established mouse model for sensory symptoms in FXS and ASD[25]. We recorded the activity of individual neurons of the primary somatosensory cortex (S1) processing touch-related sensory information. We focused on paw-related tactile sensory information given that this sensory modality is directly translational due to its high comparability between humans and mice[26]. Moreover, touch is one of the most frequently affected sensory modalities in FXS and autism[1–3], and the earliest sense to develop, providing a vital means for children to explore the world and exchange social contact with parents and caregivers[27,28]. Our approach enabled us to provide a detailed picture of atypical tactile sensory information processing in the neocortex and to identify changes in key properties of cellular and network function. We then explored to what degree these alterations contribute to enhanced endogenous neural noise, and the link between endogenous noise and variability in sensory information processing.

## Results

To explore whether the complex features of atypical sensory information processing described in clinical studies can be recapitulated in a preclinical model, we measured the processing of tactile sensory information in individual neurons of the primary somatosensory (S1) cortex in anesthetized male *Fmr1*[−/y] mice. Tactile stimuli were given to the contralateral hindpaw (HP; Fig. 1A) or forepaw (FP; see below). Pyramidal neurons were whole-cell recorded from layer (L) 2/3 (Fig. 1A) —a neuron type that controls the gain of sensory-evoked responses[29] and that is preferentially affected in ASD[30].

### Increased trial-by-trial sensory response variability in *Fmr1*[−/y] mice

First, we explored whether the increased trial-by-trial neural variability observed in human studies[6] is also a hallmark of sensory responses in *Fmr1*[−/y] mice. Thus, we compared tactile HP stimuli-evoked excitatory postsynaptic potential (EPSP) responses to 40 repetitions of the same stimulus in S1-HP neurons of *Fmr1*[−/y] and WT littermate mice (Fig. 1B, C for examples). As expected, WT neocortical neurons responded with a typical level of trial-by-trial variability to repeated stimuli (Fig. 1B–E;[31,32]). In contrast, this variability was markedly elevated in *Fmr1*[−/y]–L2/3 pyramidal neurons. It was increased for both the amplitude (Fig. 1D; $P < 0.01$), half-width (Fig. 1E; $P < 0.01$), and slope (Supplementary Table S1) of the EPSPs, thus affecting the magnitude as well as temporal features of sensory information processing from trial to trial. Due to this greater trial-by-trial variability (i.e., noise), the signal-to-noise ratio (SNR[6]) of the EPSP amplitudes was below that of WT neurons (Fig. 1F, $P < 0.05$). Given that this variability is likely shared across many neurons of the S1–L2/3 network[33], the reliability of neocortical sensory information processing might be severely constrained in these mice.

In addition, the range of EPSP amplitudes evoked by HP stimulation (Fig. 1G), and the mean value (Fig. 1H, $P < 0.001$) were both greater in *Fmr1*[−/y] neurons, significantly increasing the possible outcome of tactile stimulus-evoked responses in these neurons. The larger mean amplitude was also accompanied by a steeper slope (Fig. 1I, $P < 0.05$)

and a faster onset of the responses following HP stimulation in *Fmr1*[−/y] neurons (Fig. 1J, $P < 0.001$). Moreover, the EPSPs were prolonged (increased EPSP half-width, Fig. 1B, K, $P < 0.001$), indicating a broader temporal window for synaptic integration[34].

Our findings demonstrate that tactile stimuli elicit highly variable responses within a larger amplitude range and temporal window within the somatosensory L2/3 network in *Fmr1*[−/y] mice. These results thus replicate the clinical phenotype of trial-by-trial variability of physiologically measured sensory responses, and could provide an explanation for the temporal sensory processing issues in autism[18,19].

### AP onset variability worsens the temporal precision of sensory processing

The aforementioned EPSP alterations suggest that the onset of tactile stimulus-evoked APs would also be more variable from trial to trial in *Fmr1*[−/y] neurons. Tactile HP stimulation elicited APs in at least some of the stimulus trials in about ~25% of the recorded neurons (Fig. 1L; proportion *Fmr1*[−/y] vs. WT, n.s.). We probed AP onset variability by measuring the latency of the first evoked AP within each trial (Fig. 1M–P). While the mean AP onset latency was not different from WT neurons (Fig. 1O, n.s.), the trial-by-trial variability was significantly increased in *Fmr1*[−/y] neurons (Fig. 1P, $P < 0.01$). An increased AP onset jitter would further reduce the coordinated and reliable processing of sensory information within neocortical circuits, contributing to the temporal processing issues in autistic individuals[18]. In addition to variable timing, we also found a ~twofold increase in the fraction of stimulus trials evoking APs in the *Fmr1*[−/y] neuronal population, thereby affecting the number of activated neurons per stimulus (Fig. 1Q–T). We conclude that any given tactile HP stimulus elicits variably timed AP firing in approximately twice as many L2/3 pyramidal neurons within the S1-HP region in *Fmr1*[−/y] mice.

To extend these findings to a well-established behavioral biomarker of atypical sensory responsiveness in *Fmr1*[−/y] mice, we re-examined previously obtained data[35] to explore the inter-trial variability of the body startle response to mild acoustic stimuli (Fig. 1U). We were surprised to find a greater response variability across trials (Fig. 1V, $P < 0.05$), supporting a possible link between neuronal variability and behavioral variability.

### Endogenous neural noise correlates with trial-by-trial variability

Endogenous noise[21,22,36] at the level of individual neurons or small neural networks has been theorized to be at the root of the variable neural sensory processing in ASD[13,16,23,24]. Our high-resolution experimental data in mice enables us to directly test the validity of this hypothesis in *Fmr1*[−/y] mice[23]. We thus probed changes in the fluctuation of the membrane potential ($V_m$ variance) as a key indicator of endogenous noise (ref. [24]; Fig. 2A, B). To better assess the impact of this noise source on sensory processing, we calculated $V_m$ variance just before the arrival of the HP stimulus-evoked response. This baseline $V_m$ variance was on average ~twofold larger in *Fmr1*[−/y] neurons (Fig. 2B, C, $P < 0.05$), supporting the high-noise model of autism[16]. If this element of endogenous noise could be the cause of the trial-by-trial variability of sensory responses, then it should similarly vary from trial to trial in a correlated manner. We confirmed this prediction by demonstrating a significantly greater trial-by-trial variability of $V_m$ variance (SD of $V_m$ variance) in *Fmr1*[−/y] neurons (Fig. 2D, E, $P < 0.05$). Importantly, the magnitude of $V_m$ variance strongly correlated with both the EPSP amplitude and EPSP half-width on a trial-by-trial basis (Fig. 2F, G; see also $V_m$ variance and EPSP amplitude for same trials, Figs. 1C and 2D).

These results implicate endogenous neural noise (in particular expressed as $V_m$ variance) as a crucial factor of variable sensory processing in *Fmr1*[−/y] neurons. The large trial-by-trial variability of endogenous noise levels results in fluctuating functional neocortical states with ensuing consequences for incoming sensory inputs.

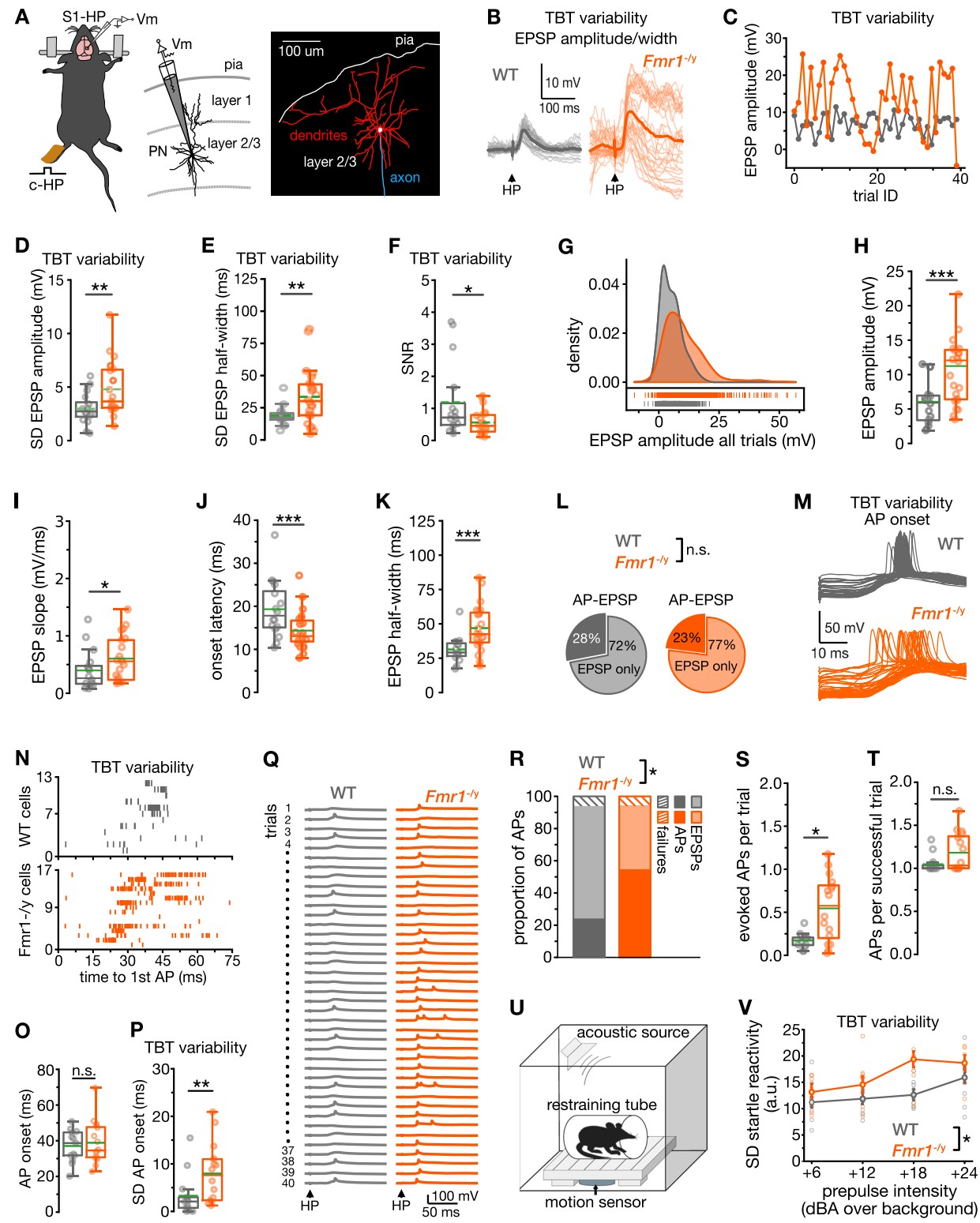

## Increased trial-by-trial variability of oscillatory power

We then explored the network components of endogenous noise and their impact on sensory processing. Network oscillations resulting from structured synaptic input patterns reflect processes linked to information transfer, perception, cognition, and behavior[37,38], and dysfunction in these oscillations has been strongly implicated in FXS and ASD, serving as physiological biomarker of altered brain states[39]. We probed whether we can detect alterations in the power of the *Fmr1*[-/y]−S1 network oscillations in our single-neuron recordings. We

analyzed the spectral power of single-neuron $V_m$ fluctuations for the commonly used frequency bands: delta (1−4 Hz), theta (4−8 Hz), alpha (8−12 Hz), beta (12−30 Hz), and gamma (30−100 Hz) (Fig. 2H−J). This analysis revealed that the power-spectral density (PSD) for this frequency range was increased in *Fmr1*[-/y] compared to WT neurons (Fig. 2H, I).

Since periodic activity directly contributes to the $V_m$ variance and thus endogenous noise, we predicted that the PSD value over this frequency range (<100 Hz) would similarly vary on a trial-by-trial

**Fig. 1 | Trial-by-trial variability of sensory responses is markedly increased in S1–L2/3 pyramidal neurons of *Fmr1*<sup>−/y</sup> mice.** A Experimental setup (left) and morphological reconstruction of a recorded neuron (right). B–K Analysis for $n = 24$ cells from 14 *Fmr1*<sup>−/y</sup> mice and $n = 16$ cells from 7 WT mice. B–F Trial-by-trial (TBT) variability of EPSPs. B Example TBT variability of EPSP amplitudes across 40 trials for one WT and one *Fmr1*<sup>−/y</sup> cell (darker color, average response). C Same traces, plotted versus number of trials. Standard deviation (SD) of EPSP amplitudes (D) and half-width (E) across trials. F Signal-to-noise ratio (SNR) of responses (EPSP amplitude divided by response variance across trials). G Density plot showing distribution of EPSP amplitude values. Box plots of EPSP amplitude (H), rise slope (I), onset latency (following stimulus) (J), and half-width (K). L Pie charts showing the proportion of neurons responding with APs in some of the trials (AP-EPSP neurons), or EPSPs only (EPSP-only neurons). M–P TBT onset variability of AP onset following HP stimulation. (AP-EPSP neurons; *Fmr1*<sup>−/y</sup>, $n = 16$ cells from 15 mice; WT, $n = 14$ cells from 12 mice). M Example traces of HP stimulus-evoked AP responses. N Onset of each first AP across 40 HP-stimulation trials indicated for all AP-EPSP neurons. O Trial-wise average of AP delay. P TBT AP jitter. Q–T Properties of HP stimulus-evoked APs. Q Example of responses to 40 successive HP stimuli. R Stacked bar graph showing percentage for each response outcome (APs, EPSPs, failures) to HP stimuli. S Box plots showing number of evoked APs averaged across all 40 trials. T Number of APs per successful trial (i.e., AP evoking trial). U Acoustic startle test. V TBT variability of responses. WT $n = 10$ mice; *Fmr1*<sup>−/y</sup> $n = 10$ mice. Statistical significance was calculated using two-sided unpaired *t* test (J, O), Fisher's exact test (L), two-sided Chi-square 3 × 3 test (R), two-sided Mann–Whitney *U* test (D, E, F, H, I, J, K, P, S, T) or repeated measure two-way ANOVA (V). n.s., not significant, *$P < 0.05$, **$P < 0.01$, ***$P < 0.001$.

basis during the baseline phase just prior to the incoming sensory response. Indeed, the trial-by-trial variability of the baseline PSD was significantly increased in *Fmr1*<sup>−/y</sup> neurons (Fig. 2J, $P < 0.01$), and correlated with both the EPSP amplitude and duration on a trial-by-trial basis (not shown). Our data at the single-cell level show that dysfunction in oscillatory synaptic input patterns provide an important endogenous noise source for unreliable sensory processing. These results also suggest that measures of oscillation power in clinical studies could be used to predict endogenous neural noise and trial-by-trial variability in neocortical processing in both FXS and autism.

### Greater variability in up–downstate difference

Neocortical states transition between quieter, hyperpolarized downstates, and more active, depolarized upstates[40] that are characterized by the presence of synaptic input patterns with high-frequency oscillatory components (Fig. 2K). Sensory responses evoked during either an up- or a downstate will be shaped by differences in membrane impedance and synaptic driving force[41] (example cells, Fig. 2L). In line with our findings of increased trial-by-trial sensory response variability, our fine-scale analysis of the neocortical state transitions revealed that upstates were often fractionated by brief "micro"-upstates (100–150 ms) in *Fmr1*<sup>−/y</sup> neurons (Fig. 2M, N). The presence of micro upstates (Fig. 2O, $P < 0.01$) was coupled with an overall increase in the upstate frequency (Fig. 2P, $P < 0.01$), and accompanied by a reduced duration and increased frequency of downstates (Supplementary Table S1; $P < 0.01$ for both). In addition to the temporal dynamics, the up–downstate $V_m$ difference was significantly larger for the *Fmr1*<sup>−/y</sup> group (Fig. 2Q, $P < 0.05$). Moreover, we observed a greater range of upstate $V_m$ values (normalized to downstate $V_m$) in *Fmr1*<sup>−/y</sup> neurons (Fig. 2R). Altogether, these alterations would create a broader range of synaptic driving force and membrane impedance for incoming sensory responses, in turn contributing to a larger range of EPSP amplitudes (Figs. 1G and 2S). Our data suggest that the increased up–downstate $V_m$ difference and upstate $V_m$ range present crucial endogenous noise features driving increased sensory response variability in *Fmr1*<sup>−/y</sup> neurons.

### Elevated spontaneous activity in *Fmr1*<sup>−/y</sup> neurons

In S1, action potentials (APs) are preferentially evoked during upstates, and spontaneous AP activity strongly impacts on sensory information processing (reviewed in ref. 42). A significant change in this feature might thus indicate a background noise element of the S1 circuitry, altering its operation[21,43]. We found that a larger fraction of the *Fmr1*<sup>−/y</sup> neuronal population was spontaneously active (Fig. 3A, $n = 4/16$ WT and $8/17$ *Fmr1*<sup>−/y</sup> neurons, $P < 0.01$). In addition, while spontaneous AP activity in WT neurons was characteristically low[44,45] (~0.005 Hz), *Fmr1*<sup>−/y</sup> neurons exhibited a significantly increased spontaneous AP frequency (Fig. 3B, C, $P < 0.05$). These findings suggest a more active basal S1 network state in *Fmr1*<sup>−/y</sup> mice.

### L2/3 pyramidal neurons are more excitable in *Fmr1*<sup>−/y</sup> mice

We next explored the contributory role of altered intrinsic excitability to endogenous noise. Our data revealed that the number of APs and the maximum AP frequency generated by depolarizing current steps were significantly increased in *Fmr1*<sup>−/y</sup> S1-HP neurons (Fig. 3D, E, $P < 0.05$), demonstrating that they are intrinsically more excitable. In addition, the APs were wider (Fig. 3F, G, $P < 0.05$) and the after-depolarization (ADP) amplitude of the membrane potential following brief high-frequency bursts of APs was increased (Fig. 3H, $P < 0.05$). The former feature suggests an increased likelihood of AP-evoked transmitter release at the neuron's axon terminals[35,46], whereas the latter suggests a dendritic hyperexcitability phenotype[35,47], both acting together to amplify the input–output function of these neurons and ultimately the spread of excitation within the neocortex. Altered excitability might also contribute to the endogenous cellular and circuit noise of S1 by elevating the spontaneous AP firing and increasing the E/I ratio within S1[13], thus rendering sensory information processing less reliable. Other intrinsic properties, such as resting membrane potential, rheobase, and AP threshold, remained unaltered (Supplementary Table S1).

### *Fmr1*<sup>−/y</sup> neurons have exaggerated variability in numerous features

Noisy sensory processing in autistic individuals is characterized not only by increased trial-by-trial variability but also by greater inter-individual variability[48]. We tested changes in the variability across our neuronal population as a readout for variability within the *Fmr1*<sup>−/y</sup> neuronal population. We compared the variability of both noise- and sensory response related parameters (Supplementary Table S2). This analysis demonstrates that the variance of sensory stimulus-evoked EPSP (e.g., EPSP half-width, Fig. 3I, $P < 0.05$) and AP (e.g., AP number, Fig. 3J, $P < 0.001$) responses were higher for the *Fmr1*<sup>−/y</sup> population than that for the WT population. In addition, variance was also increased for a number of core noise features, including $V_m$ variance (variance of $V_m$ variance, Fig. 3K, $P < 0.01$), oscillation power (e.g., gamma-band power, Fig. 3L, $P < 0.05$), spontaneous AP firing (Fig. 3M, $P < 0.001$), and intrinsic excitability (see Supplementary Table S2 for complete list). These results suggest the presence of a greater diversity of functional S1 network states in *Fmr1*<sup>−/y</sup> mice. In, we found an increase in the inter-individual variability of the behavioral startle response (Figs. 1U and 3N), supporting the idea that elevated inter-individual variability is a hallmark of altered sensory neocortical processing in ASD.

### Broader tuning of S1 neurons suggests reorganization of connectivity

Noisy neural circuits impact the precision of sensory processing and, together with anatomical-functional alterations, affect the connectivity of neocortical neurons—changes in which have been described in both individuals with autism and ASD mouse models[48–51]. We

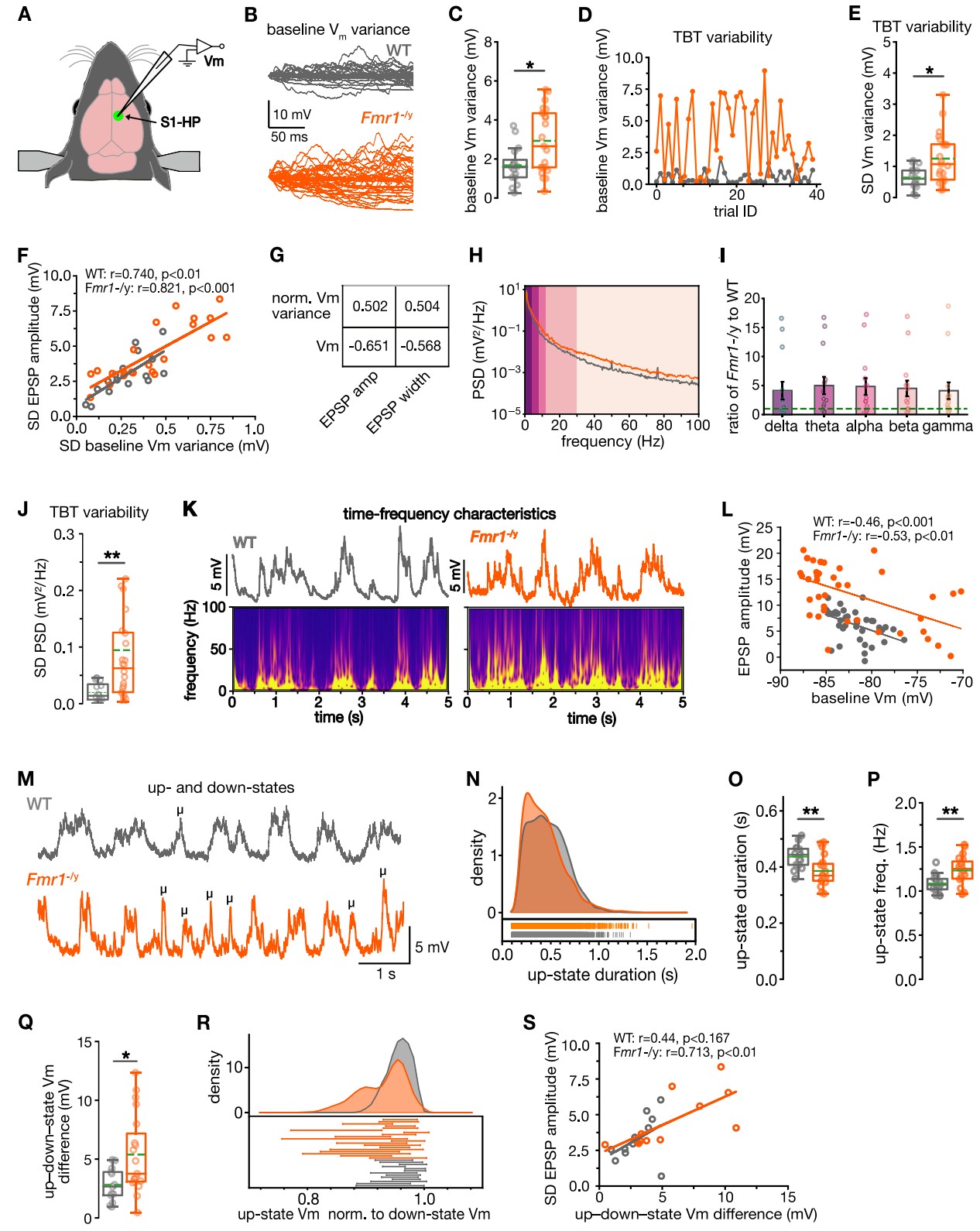

tested the relevance of these changes for the specificity of sensory processing, by measuring the receptive field properties of layer 2/3 pyramidal neurons of S1-HP in *Fmr1*-/y mice. Specifically, we asked whether HP neurons would display differential responsiveness to tactile forepaw (FP) stimulation (Fig. 3O–Q). Our results indicate that the percentage of neurons responding to both HP and FP stimuli shifted from ~20% in WT mice to ~50% in *Fmr1*-/y mice (Fig. 3Q, *P* < 0.05). This

finding suggests a broader and less specific tuning of these neurons to tactile stimuli of different sub-modalities.

## Relationship between endogenous noise and atypical sensory information processing

To integrate the aforementioned findings into a functional model, we developed a correlation matrix (Fig. 4A, B) and visualized the

**Fig. 2 | Endogenous neural noise drives trial-by-trial variability of sensory responses. A** Experimental setup. **B–J** Fluctuation of the membrane potential ($V_m$ variance). (*Fmr1$^{-/y}$*, $n = 23$ cells from 14 mice; WT, $n = 16$ cells from seven mice) **B** Example for baseline (200-ms time window just before the onset of the HP stimulus) $V_m$ variance. **C** Baseline $V_m$ variance. **D** Same trials as shown in (**B**) and Fig. 1C, showing intra-cell variability and correlation of baseline $V_m$ variance with EPSP amplitude. **E** TBT variability in baseline $V_m$ variance (SD of baseline $V_m$ variance). **F** For each neuron, the SD of EPSP amplitudes across 40 trials was plotted vs the SD of baseline Vm variance (normalized by the Vm). **G** Average correlation coefficients from all the WT and *Fmr1$^{-/y}$* neurons data pooled together (**H–J**) PSD (*Fmr1$^{-/y}$*, $n = 23$ cells from 14 mice; WT, $n = 16$ cells from 7 mice). **H** Power-spectral density (PSD) across a frequency spectrum of 0–100 Hz. **I** *Fmr1$^{-/y}$*-to-WT PSD ratio for different frequency bands. **J** TBT variability of baseline PSD. **K–S** Up–downstate

dynamics and EPSP–Vm relationship (*Fmr1$^{-/y}$*, $n = 19$ cells from 16 mice; WT, $n = 13$ cells from eight mice). **K** Example of power–time–frequency characteristics of $V_m$ calculated using wavelet transformation. **L** Correlation between EPSP amplitude and baseline Vm for a *Fmr1$^{-/y}$* and a WT neuron. **M** Example traces of up–downstate dynamics; short-duration (100–150 ms) upstates are marked with **μ**. **N** Density distribution histogram of upstate duration. Box plots of upstate duration (**O**), upstate frequency (**P**), and up–downstate Vm difference (**Q**). **R** Density distribution histogram of upstate $V_m$ normalized to mean downstate Vm. **S** SD of EPSP amplitudes correlates with up–downstate Vm difference. Statistical significance was calculated using two-sided unpaired $t$ test (**O, P** and **Q**), two-sided Mann–Whitney $U$ test (**C, E, J**), two-sided Pearson $R$ test (**F, L, S**). n.s. not significant, *$P < 0.05$, **$P < 0.01$.

statistically significant correlations of the most relevant noise and sensory responses features in the form of a node graph (Fig. 4C, D). This allows us to evaluate the predictive value of our physiological measures for cellular or network function with potential links to increased variability of cellular responses in autism. This analysis revealed that the most prominent nodes based on their number of significant correlations with other parameters were: $V_m$ variance, up–downstate $V_m$ difference, oscillation power, trial-by-trial variability, and EPSP amplitude.

The main findings from the correlation matrix can be summarized as follows: (i) For both *Fmr1$^{-/y}$* and WT neurons, the (baseline) $V_m$ variance was positively correlated with EPSP amplitude, measures of trial-by-trial (tbt) variability, and the power of theta- and alpha oscillations, and negatively with SNR. For *Fmr1$^{-/y}$* neurons, the baseline $V_m$ variance was also positively correlated with the power of delta and gamma oscillations, while this correlation was absent in WT neurons. These findings suggest that $V_m$ variance represents an important driver of sensory response amplitude, and in particular its trial-by-trial variability. (ii) For *Fmr1$^{-/y}$* neurons, the up–downstate $V_m$ difference was positively correlated with the EPSP amplitude, baseline $V_m$ variance, and trial-by-trial measures of baseline $V_m$ variance, PSD, and EPSP amplitude. In contrast, these correlations were not present in WT neurons. (iii) The power of the oscillatory bands strongly correlated with several parameters belonging to different categories in *Fmr1$^{-/y}$* neurons compared with WT neurons. In particular, there was a strong positive correlation between the power of several oscillations and trial-by-trial variability measures. In addition, gamma power was positively correlated with spontaneous AP firing, baseline $V_m$ variance, and upstate frequency. (iv) For *Fmr1$^{-/y}$* neurons, the spontaneous AP activity positively correlated with the beta and gamma powers. (v) For *Fmr1$^{-/y}$* neurons, the maximal AP firing rate (intrinsic excitability measure) was positively correlated with the EPSP half-width and peak latency, and negatively correlated with the beta power. The 3rd AP half-width was positively correlated with the EPSP onset latency and trial-by-trial EPSP half-width.

Altogether, for the *Fmr1$^{-/y}$* neuronal population many more parameters correlated positively with each other when compared to the WT neurons. Among these parameters, the strongest correlations were found between endogenous noise sources, variability measures and EPSP parameters. $V_m$ variance, up–downstate $V_m$ difference, and oscillation power emerge as core endogenous noise parameters that strongly determine atypical sensory information processing in the S1 network. Thus, trial-by-trial variability of sensory responses is largely attributable to the neocortical state at the time of the incoming sensory responses. More specifically, *Fmr1$^{-/y}$* neurons displaying the strongest HP stimuli-evoked responses are also those exhibiting the largest trial-by-trial variability, $V_m$ variance and up–downstate $V_m$ difference. This correlation between the three aspects of information processing is highly pertinent and could point to important physiological biomarkers for clinical studies.

## Dissecting the origin of the different endogenous noise sources

We asked whether we could further dissect the origin and functional role of the different components of endogenous noise in *Fmr1$^{-/y}$* neurons. To address this, we pharmacologically targeted the voltage-, and calcium-sensitive K$^+$ channel, BK$_{Ca}$ channel, an approach that has previously been shown to correct cellular hyperexcitability[35,46,52,53]. To test whether the endogenous noise elements and atypical sensory responses can be modulated by pharmacological manipulation at the level of the S1, we developed an in vivo neocortical assay in which we applied a BK$_{Ca}$ channel agonist locally to the S1 surface (Fig. 5A). This strategy was enabled by employing the highly selective BK$_{Ca}$, agonist, BMS191011[54], which has poor penetration in complex tissue such as the brain due to its binding to protein and other complex biomolecules. Our strategy allowed us to distinguish cellular and network deficits that were sensitive to local manipulation of BK$_{Ca}$ channels from those that were not affected.

We found that local S1 application of BMS191011 was surprisingly effective in rescuing physiopathological alterations in *Fmr1$^{-/y}$* neurons. Many features of HP stimulus-evoked EPSPs, including the mean amplitude, half-width, and onset latency, were rescued by BMS191011 application (Fig. 5B–E; WT vs. *Fmr1$^{-/y}$* + BMS191011; all n.s.). In addition, while trial-by-trial variability of EPSP amplitude was not affected by local BMS191011 application in *Fmr1$^{-/y}$* neurons (Fig. 5F, $P < 0.01$), that of the EPSP half-width and EPSP slope was not significantly different from the WT group after treatment (Fig. 5G, H, both n.s.).

Accordingly, several alterations related to endogenous noise features were not different from WT values following BMS191011 treatment, including baseline $V_m$ variance, spontaneous AP firing (also percentage of spontaneously active cells, Supplementary Table S1), and AP half-width (Fig. 5I–M; WT vs. *Fmr1$^{-/y}$*-BMS191011; all n.s.). In contrast, up–downstate $V_m$ difference (Fig. 5N, $P < 0.01$), and the power of most oscillations (Supplementary Table S3) were not rescued.

Moreover, BMS191011 diminished the inter-neuronal variability of several features related to sensory responses; for example, the EPSP amplitude variability was reduced below that of WT neurons (Fig. 5O, $P < 0.05$), while there was no significant difference between WT and *Fmr1$^{-/y}$* values for EPSP half-width and spontaneous AP firing rate following pharmacological treatment (Fig. 5P–Q, both n.s.). Finally, at the behavioral level, i.p. injection of a different BK$_{Ca}$ channel agonist with high blood–brain–barrier permeability, BMS204352, rescued trial-by-trial variability (Fig. 5R, n.s.) as well as inter-individual variability (Supplementary Table S3) of the acoustic startle response.

Collectively, our results show that localized BMS191011 application reduces elevated spontaneous AP firing and neuronal excitability, dampens the impact of HP stimulus-evoked responses, and reduces $V_m$ variance and variability in S1–*Fmr1$^{-/y}$*–L2/3 pyramidal neurons. Rescue of these features could be explained by the role of BK$_{Ca}$ channels in regulating AP properties and thus neurotransmitter release properties at local S1 synapses, dendritic excitability and integration of S1 neurons, as well as their AP output. Notably, BMS191011 application within

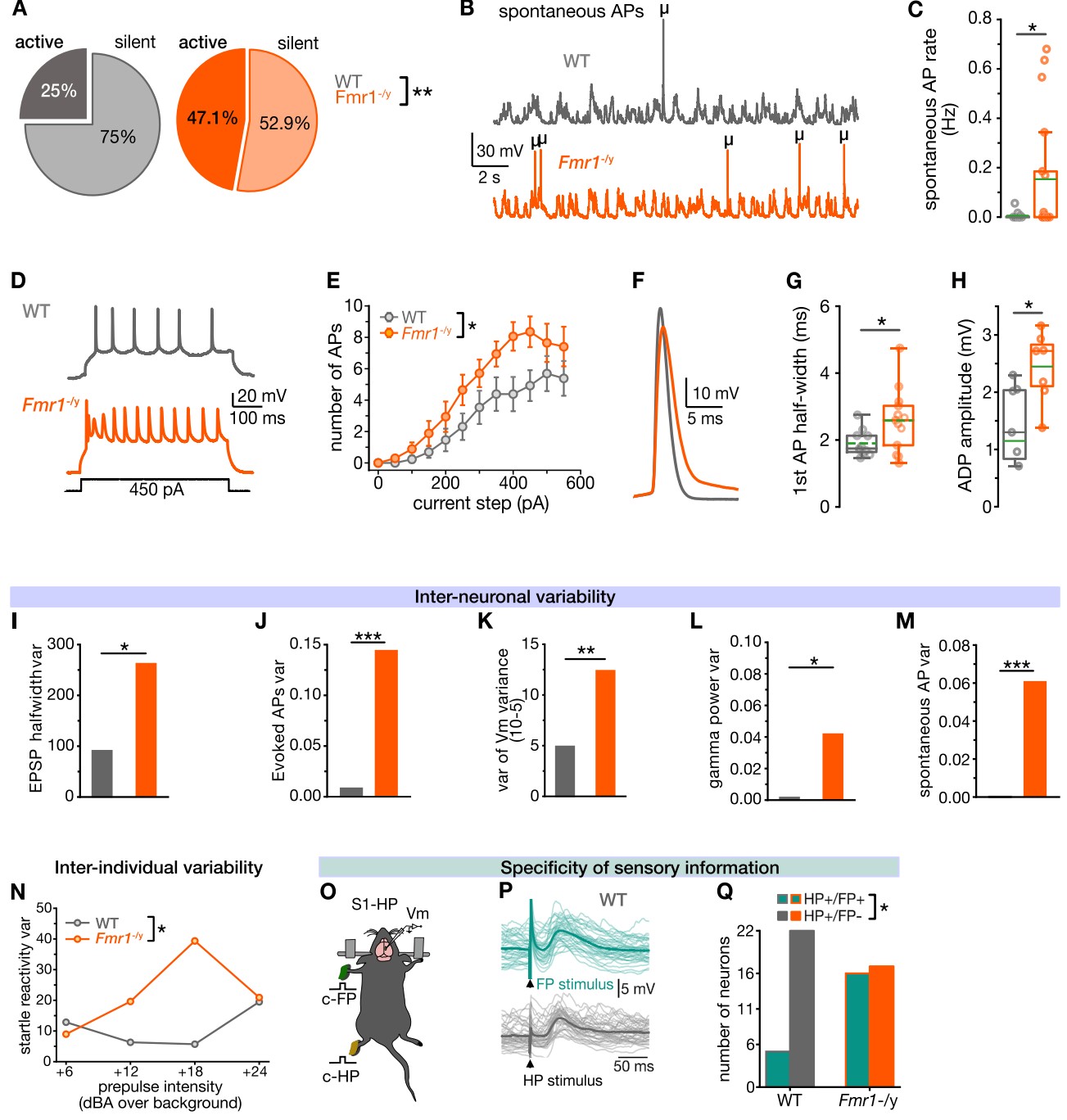

**Fig. 3 | Changes in action potential properties, inter-neuronal variability, and receptive field specificity in *Fmr1⁻/y* mice. A–C** Spontaneous AP activity (*Fmr1⁻/y*, 17 cells from 15 mice; WT, 16 cells/10 mice). **A** Pie charts showing the percentage of neurons that were spontaneously active (dark color) or silent (light color) during a time window of 120 s. **B** Representative example traces; μ-symbols indicate spontaneous APs. **C** Box plot showing AP frequency. **D–H** Intrinsic properties. **D** Example voltage traces in response to a depolarizing current step. **E** Mean number of APs plotted as function of the injected current (*Fmr1⁻/y*, 17 cells/16 mice; WT, *n* = 13 cells/8 mice). **F** Example traces showing the broadening of APs. **G** Box plot showing the first AP half-width (*Fmr1⁻/y*, 15 cells/11 mice; WT, 11 cells/7 mice). **H** Average amplitude of after-depolarization (ADP) (*Fmr1⁻/y*, 7 cells/3 mice; WT, 7 cells/2 mice). **I–M** Inter-neuronal variability across cell population. Bar graphs illustrating variance of EPSP half-width (*Fmr1⁻/y*, 24 cells/14 mice; WT, 16 cells/7 mice) (**I**), HP

stimulus-evoked APs (*Fmr1⁻/y*, 16 cells/14 mice; WT, 14 cells/12 mice) (**J**), Vₘ variance (*Fmr1⁻/y*, 23 cells/14 mice; WT, 16 cells/7 mice) (**K**), gamma power (*Fmr1⁻/y*, 24 cells/14 mice; WT, 16 cells/7 mice) (**L**), and of spontaneous APs (*Fmr1⁻/y*, 17 cells/15 mice; WT, 16 cells/10 mice) (**M**). **N** Inter-individual variability of responses in the acoustic startle test (10 *Fmr1⁻/y* mice and 10 WT mice). **O–Q** Receptive field properties of S1-HP L2/3 pyramidal neurons (*Fmr1⁻/y*, 33 cells/28 mice; WT, 27 cells/23 mice). **O** Experimental schematic. **P** Examples for HP and FP stimulus-evoked responses in a WT neuron. **Q** Number of neurons responding to HP–only stimulus and to HP/FP stimuli. Statistical significance was calculated using two-sided Chi-square (**A**, **P**), two-sided unpaired *t* test (**G**, **H**), or two-sided mixed ANOVA (**E**), two-sided Bartlett variance test (**I**), two-sided Levene variance test (**L**, **K**, **M**), F test (**J**), Two-sided permutation test (**C**), two-sided Fisher's exact test (**Q**). n.s. not significant, *P < 0.05, **P < 0.01.

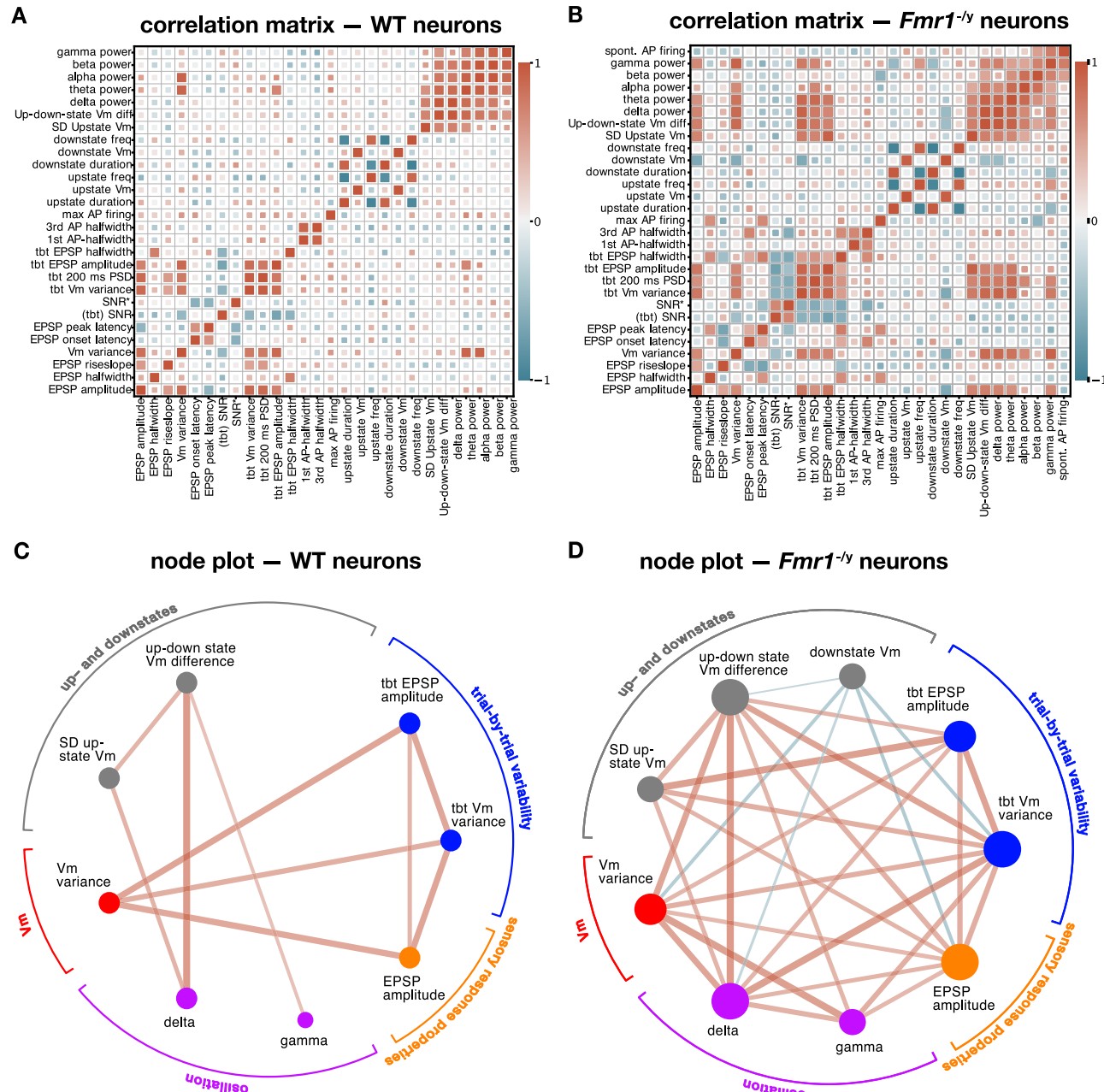

**Fig. 4 | Stronger correlation of endogenous noise with sensory response variability in *Fmr1⁻/y* neurons. A**, **B** Correlation of main parameters describing spontaneous AP firing, up-/downstate pattern, intrinsic excitability, power-spectral−density, HP stimulus-evoked responses and trial-by−trial (tbt) variability. Positive correlations are indicated in red and negative correlations in blue. The correlation strength is color-coded, and statistically significant correlations are indicated by large squares. Correlation matrix of WT neurons (**A**), and *Fmr1⁻/y*

neurons (**B**). **C**, **D** Node graphs displaying only statistically significant correlations of the main parameters related to endogenous noise and sensory response variability. The size of the node is proportional to the number of significant correlations with other parameters. The thickness of the edges is proportional to the coefficient (strength) of the correlation. The parameters have been grouped under the same color according to their biological similarities. Node graph of WT (**C**), and *Fmr1⁻/y* (**D**) neurons.

S1 did not rescue the trial-by-trial variability of EPSP amplitudes, nor the power of periodic synaptic input patterns or up−downstate $V_m$ difference.

Our results provide evidence for the usefulness of mechanism-based targeted approaches for the dissection of the different noise sources, and the examination of their relationship with atypical sensory processing features.

## Discussion

These results demonstrate that many of the complex variability features of sensory processing observed in clinical studies are

recapitulated with surprising fidelity in a preclinical mouse model of FXS and autism. Crucially, we discovered core endogenous noise elements that are strongly correlated with elevated response variability. These results suggest that this elevated response variability has both cellular and network origins. Our work thus provides a framework for understanding the role of endogenous noise in atypical sensory information processing in neurodevelopmental disorders such as autism and FXS. With this in mind, we developed a model integrating the principal sources and features of altered endogenous neural noise and their contribution to atypical variability and unreliability of sensory information processing (Supplementary Fig. S1).

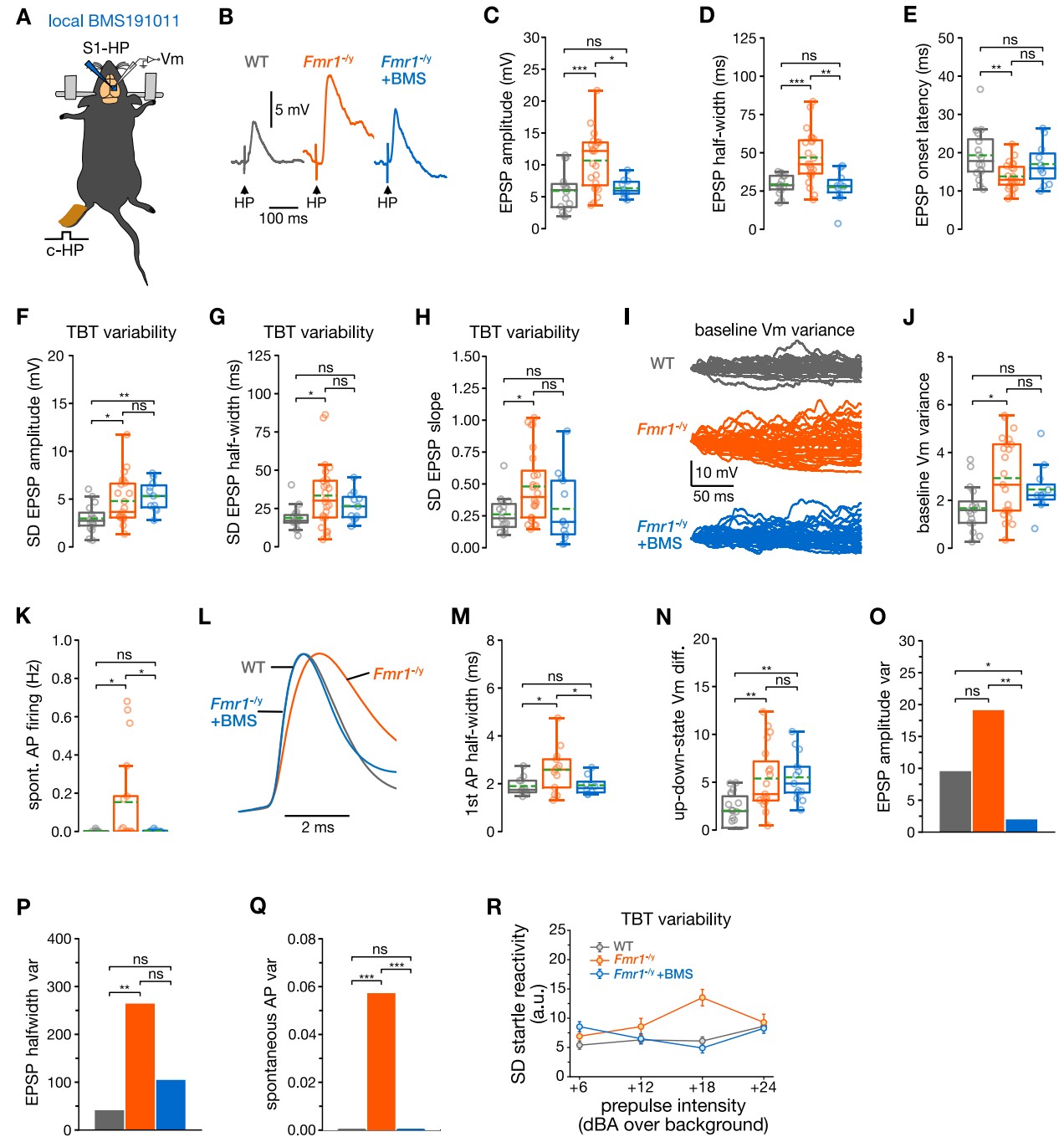

**Fig. 5 | Dissecting the origin of noise sources by targeting local neocortical ion channel dysfunction. A** Schematic of local BMS191011 (BMS) application onto S1, combined with whole-cell recordings. **B** Example traces of HP stimulus-evoked EPSPs from a WT−, *Fmr1*−/y− and a *Fmr1*−/y neuron in the presence of BMS (*Fmr1*−/y−BMS neuron) revealing correction of EPSP amplitude and half-width by BMS191011. **C–E** EPSP features (*Fmr1*−/y + BMS, 10 cells/10 mice). Box plots showing correction of EPSP amplitude (**C**), EPSP half-width (**D**), and EPSP onset latency (**E**). **F–H** TBT variability of EPSP features. **F** Lack of correction of TBT variability of EPSP amplitude (WT, 16 neurons; *Fmr1*−/y, 23 neurons; KO + BMS, 11 neurons). Correction of TBT variability of EPSP half-width (**G**) (WT, 15 neurons; *Fmr1*−/y, 24 neurons; KO + BMS, 11 neurons). **H** EPSP slope (WT, 15 neurons; *Fmr1*−/y, 24 neurons; KO + BMS, 13 neurons). **I–N** Endogenous noise features and sources. **I** Example traces of baseline Vm variance. **J** Correction of atypical baseline Vm variance by BMS191011 (WT, 16 neurons; *Fmr1*−/y, 23 neurons; KO + BMS, 11 neurons). **K** Correction of spontaneous AP firing (*Fmr1*−/y−BMS, 12 cells/12 mice). **L** Example traces of APs

demonstrating correction of 1st AP half-width by BMS191011. APs were scaled to the peak to visualize differences in half-width. **M** Box plot showing correction of AP half-width (*Fmr1*−/y−BMS, 8 cells/8 mice). **N** Lack of correction of up−downstate Vm difference by BMS1910115 (WT, 13 neurons; *Fmr1*−/y, 19 neurons; *Fmr1*−/y + BMS, 13 neurons). **O–Q** Inter-neuronal variability. **O** Bar graph demonstrating reduction in inter-neuronal variability of EPSP amplitude below that of WT neurons. Correction of inter-neuronal variability of EPSP half-width (**P**) and of spontaneous AP firing (**Q**). **R** Correction of TBT variability of acoustic startle response following i.p. injection of BMS204352 (WT, 10 mice; *Fmr1*−/y, 10 mice; *Fmr1*−/y + BMS, 8 mice). *Fmr1*−/y + BMS, *Fmr1*−/y, and WT values were statistically compared. Statistical significance was calculated using one-way ANOVA with Dunn's multiple comparisons test (**C**, **D**, **E**, **F**, **G**, **H**, **J**, **N**), Bonferroni's multiple comparisons test (**K**, **M**), two-sided permutation test (**G**), Bartlett test (**O**, **P**), or Levene test (**Q**). n.s. not significant, *P < 0.05, **P < 0.01, ***P < 0.001.

We propose that changes in network-level synaptic inputs (synaptic noise) impinging on S1–$Fmr1^{-/y}$–L2/3 pyramidal neurons together with dysfunction in their intrinsic excitability (ion channel noise) give rise to elevated endogenous noise. At the single-cell level, this endogenous noise is expressed as alterations in the oscillatory power, up–downstate difference, and variance of the membrane potential, as well as an elevated level of spontaneous AP firing as core noise components. These features are strongly correlated with the variability of sensory responses, in particular changes in their magnitude and temporal aspects. In addition, these noise components contribute to, and influence each other, further exacerbating their impact on sensory processing. For instance, an increased power in network oscillations directly increases $V_m$ variance; $V_m$ variance in turn initiates up–downstate transitions, which together with the greater up–downstate $V_m$ difference drives a larger range and trial-by-trial variability of EPSP amplitudes. Thus, the presence of fluctuating levels of endogenous noise underlies rapid changes in neocortical functional connectivity by creating an unstable S1 circuitry.

Within the constraints of our model (representing a monogenic, syndromic form of autism and not idiopathic autism), we speculate that these unstable functional S1 states could be expected to have manifold effects on sensory information processing in autism. Importantly, our findings could explain the greater trial-by-trial variability of sensory responses observed in autistic individuals. Our data also provide the first experimental evidence addressing the question whether endogenous noise is increased or decreased in small-scale networks in autism. Moreover, our model points to a set of translational biomarkers which may be predictive of endogenous noise and can be measured in autistic individuals.

Spontaneous AP activity in the neocortex contributes to the dynamicity of the network state, interacting with the processing of incoming sensory information[21,43], and enabling adaptive behavior[55]. An elevated level of spontaneous AP activity may impact the capacity to correctly predict, perceive, and interpret incoming information (reviewed in refs. 8,42). Our finding of increased spontaneous AP activity of S1–L2/3 neurons in $Fmr1^{-/y}$ mice is consistent with previous findings in both awake and anesthetized mice[56]. In addition, we found that a larger fraction of the $Fmr1^{-/y}$–L2/3 population was spontaneously active, suggesting an overall augmented background noise of the S1 network in $Fmr1^{-/y}$ mice. This increased background noise will influence the rhythmicity of network oscillations and contribute to the increased $V_m$ variance observed in $Fmr1^{-/y}$–S1 neurons with ensuing functional consequences for sensory information processing, as shown by our correlation analysis (Fig. 4 and Supplementary Fig. S1).

Neocortical states are characterized by periodic up–downstate transitions occurring during quiet rest, sleep, and anesthesia, as well as when animals perform perceptual tasks, and influence the responses of somatosensory neurons to subsequent sensory stimulation (reviewed in ref. 57). Upstates are associated with a marked increase in local neocortical network activity and in the power of higher-frequency component oscillations (including gamma, Fig. 2G)[58]. The heightened dynamicity of up–downstate transitions in S1-$Fmr1^{-/y}$ neurons together with the greater range of up–downstate $V_m$ differences is expected to result in a larger range of response magnitudes due to larger differences in the driving force and membrane impedance. These dynamic network state changes could conceivably contribute to heterogeneity in sensory features in autism, namely hyper- and hyposensitivity reported for the same sensory modality.

Our data indicate the presence of a higher power of delta, theta, alpha, beta, and gamma oscillations in $Fmr1^{-/y}$ neurons. Due to their contribution to endogenous neural noise and thereby sensory information processing (Fig. 4 and Supplementary Fig. S1), dysfunction in these oscillations suggests changes in processes linked to information transfer, perception, cognition, and behavior[37,38]. Notably, the augmentation in gamma power might relate to the aforementioned

increase in spontaneous AP firing in $Fmr1^{-/y}$ neurons (Fig. 4B, D;[32,59]), and reflect increased neocortical network excitation and altered E/I ratio[60,61]. Elevated broadband gamma noise is also associated with reduced spike precision and ability to synchronize periodic gamma band activity, and social and sensory processing difficulties, and may have cascading effects on cognitive, behavioral, and neuropsychiatric symptoms[62,63]. Our findings of a wide range of gamma power values among the $Fmr1^{-/y}$ neuronal population suggests that there might be subgroups of individuals with FXS based on the presence of higher and lower gamma power levels[64]. A link between increased gamma power and L2/3 network hyperexcitability has also been described ex vivo for the auditory cortex[65]. While it is feasible that ketamine might influence WT and $Fmr1$ KO mice differently, in particular with regards to the gamma power, literature suggests that a more likely mechanism for the increased gamma band could be a dysfunction of parvalbumin-positive interneurons in $Fmr1$ KO mice[66,67]. Mechanistic insight into abnormal gamma power from single-neuron recordings may have ramifications for better understanding the pathophysiology of sensory symptoms.

Our findings suggest that key parameters of endogenous neural noise, especially $V_m$ variance, up–downstate $V_m$, and oscillatory power, are altered, crucially impacting the magnitude, temporal resolution, and variability of tactile sensory responses within the L2/3 network of $Fmr1^{-/y}$ mice (Fig. 4 and Supplementary Fig. S1). Indeed, we find that the onset of EPSPs following hindpaw stimulation is faster, their amplitude and rise slope increased, and their duration prolonged. Tactile stimulation also evokes more variably timed APs and a greater number of APs within the L2/3 network which, together with broadening of APs, increases the probability of spreading this information to postsynaptic targets. Greater AP latency variability to sensory stimuli has previously also been described for auditory cortical neurons[68]. Augmentation of sensory responses along with greater variability across trials results in a larger repertoire of potential response features, including a larger amplitude range and wider temporal integration window, and APs that are consequently evoked at very different time points. This larger dynamic range strongly correlates with endogenous noise, which shows an overall increase but also significant variability on a trial-by-trial basis. An optimal endogenous noise level could be beneficial[69,70], improving neuronal responsiveness and thus perceptual detection sensitivity, whereas higher noise levels would randomize neuronal responses and impair behavioral performance[71]. Together, these variable network states have the potential to contribute to superior and inferior sensory detection and discrimination skills, sensory hypersensitivity, hyposensitivity, as well as temporal processing issues in autism[6,12,15,18,19,72,73]. In a hyper-excited state, the network's computational property of divisive normalization is degraded and it's E/I ratio is enhanced[13,74,75]. Our work suggests a complex scenario in which more variable neocortical sensory responses form part of the neurophysiological signature of FXS and autism.

In human studies, the consequences of noise present as increased variability in the magnitude and dynamics of evoked neuronal responses to sensory stimuli[7,8,11,12]. We find it intriguing and highly pertinent that these hallmarks of atypical sensory responses, measured in clinical studies at the scale of large neuronal networks (fMRI, EEG, or even behavioral responses) are recapitulated by our single-cell measures and limited behavioral measures in a preclinical model. These parallels between neural measures reflecting small-scale networks and human physiological responses suggest that it is possible to exploit our findings to dissect the mechanisms underlying complex clinical measures.

Lastly, our data demonstrate a higher prevalence of tactile fore-paw stimulus-related responses in L2/3 pyramidal neurons of the S1 hindpaw region, indicating reorganization of the structural or functional connectivity of the sensory cortex (for similar findings in the visual cortex, see ref. 49). Functionally, this results in changes in the

receptive field properties of L2/3 pyramidal neurons of S1-HP. Similar findings have been described for L2/3 pyramidal neurons of the whisker-related S1 barrel cortex[76] and auditory cortex[68]. Changes in synaptic connectivity would add noise to the network, leading to compound changes in neuronal computation with negative consequences for sensory filtering or transmission of information to associative sensory areas, and potentially behavioral or perceptual responses, as suggested by Cascio et al.[77]. Functional connectivity reorganization, along with enhanced stochastic resonance are also proposed neural mechanisms of synesthesia, which commonly occurs in ASD[14,78]. Together with the increased probability of neurotransmitter release due to broadened APs[46], the increased level of sensory stimulus-evoked APs and connectivity reorganization provide a multiplicative noise source by more efficiently spreading information to a larger postsynaptic network.

A major challenge to sensory-based autism research is the development of optimized biomarkers that allow direct comparisons between rodents and human subjects while simultaneously permitting exploration of the underlying neurobiological mechanisms. Our analysis of single neocortical neurons in $Fmr1^{-/y}$ mice suggests a change in basal oscillatory power over a range of frequencies, and could be used as a cross-species marker. Changes in basal oscillatory power strongly correlate with other sources of endogenous noise, particularly $V_m$ variance, up–downstate $V_m$ difference, and spontaneous AP firing, as well as with measures of trial-by-trial variability for sensory-evoked responses. Although a specific relationship with trial-by-trial variability has not yet been established and merits further exploration, a relationship between changes in gamma (30–80 Hz) power and sensory issues has been suggested in human studies (reviewed in refs. 62,79). This cross-species conservation of resting-state gamma power changes is remarkable, suggesting its potential as a useful marker of endogenous noise.

By targeting a relevant cellular mechanism locally within S1, we gained further insight into the underpinnings of endogenous noise and atypical sensory processing. We chose to target $BK_{Ca}$ channels because of their role in regulating AP features, neurotransmitter release probability, and dendritic excitability–features that likely contribute to endogenous noise sources and sensory response alterations. In addition, these channels have previously been implicated in autism[52], sensory information processing[53] and FXS and suggested as suitable targets for intervention[35,46,80,81].

Our data show that local neocortical application of the selective $BK_{Ca}$ agonist, BMS191011, can correct AP half-width and ADP, reduce spontaneous AP firing, and restore multiple aspects of tactile information processing in S1-HP L2/3 pyramidal neurons including the EPSP amplitude and half-width and their inter-neuronal variability, as well as some aspects of trial-by-trial variability. This finding of a localized pharmacological intervention effect supports the idea that some of the neurobiological alterations of atypical sensory information processing in autism could be rescued by targeting neocortical neurons and their presynaptic sites.

On the other hand, local $BK_{Ca}$ channel modulation had little or no significant impact on key endogenous noise measures correlating with the trial-by-trial variability of EPSP amplitude and half-width, namely up–downstate $V_m$ difference and oscillatory power. Thus, these endogenous noise features that correlate with trial-by-rial variability either originate outside of the S1 network such as synaptic noise arising from long-ranging connections, or are based on other mechanisms. Our framework allows the evaluation of local or global manipulation on endogenous noise sources contributing to atypical sensory information processing in autism.

Neural sensory information processing consists of a stimulus-specific component and noise. Based on signal detection theory, the relationship among signal, noise, and neural output in the sensory cortices can be expressed using the following mathematically defined model[15,20,82]: $O = K(S) \times (1 + N_m) + N_a$. Here, the neuronal output (O) is a function of the encoding function of the signal (K), the signal itself (S), and various noise sources, namely, multiplicative noise ($N_m$; stimulus-related) and additive noise ($N_a$; stimulus-independent). $N_m$, for example, would cause the signal to spread more widely within the network, in line with computational motifs such as reduced divisive normalization, enhanced E/I ratio model and the "intense world theory"[13,83–85]. While it is important to note that it may be difficult to discriminate between multiplicative noise, gain control and additive noise, our experimental findings provide a starting point in this direction. In particular, our data reveal atypical features in the $Fmr1^{-/y}$ neurons that would suggest both K(S), $N_m$, and $N_a$ alterations. The endogenous noise parameters described in our study ($V_m$ variance, up–downstate $V_m$ difference, spontaneous AP firing, and network oscillation power) would be crucial contributors of additive noise. In addition, these measures would also affect the gain or encoding function (altered synaptic summation and AP output). An increase in neuronal excitability would contribute to the additive noise, a modification of the gain function (enhanced neural throughput), and multiplicative noise (enhanced spread of the signal to postsynaptic targets due to increased transmitter release probability). The alterations in the receptive field properties observed in $Fmr1^{-/y}$ neurons are indicative of an enhanced functional–structural connectivity, which would strongly affect $N_m$. The functional outcome of the combined changes would vary on a trial–by–trial basis and depend strongly on the highly fluctuating endogenous noise levels with ensuing consequences for sensory processing. Given the strong correlation between atypical sensory symptoms and autism severity, future studies encompassing measures of cellular/network noise are warranted (Fig. 6). To this end, our study points to a number of biomarkers that are likely to be useful indicators of noise. Understanding the role of noise in sensory information processing may lead to new interventional strategies, whether behavioral, environmental, or pharmacological, to relieve the stress and conflict that these experiences generate.

## Methods

### Experimental design

We performed in vivo whole-cell patch-clamp recordings of neocortical neurons of the primary somatosensory cortex (S1) or the hindpaw (HP) region to examine tactile stimulus-evoked sensory processing in anesthetized mice and to probe the causal role of endogenous noise sources and parameters for atypical sensory information processing in autism. Throughout the text, we are using terms that are preferred in the autistic community and are less stigmatizing[86].

**Ethical statement.** All experimental procedures were performed in accordance with the EU directive 2010/63/EU and French law following procedures approved by the Bordeaux Ethics Committee (CE2A50) and Ministry for Higher Education and Research. Mice were maintained under controlled conditions (temperature 22–24 °C, humidity 40–60%, 12 h/12 h light/dark cycle, light on at 07:00) in a conventional animal facility with ad libitum access to food and water. All experiments were performed during the light cycle.

**Mice.** Second-generation $Fmr1$ knockout ($Fmr1^{-/y}$)[35] and wild-type littermate mice at P26–42 were used in our study. Mice were maintained in a mixed 129/Sv/C57Bl/6 J/FVB background (backcrossed 6 generations into C57Bl/6J) as described in ref. 35. Male wild-type and $Fmr1^{-/y}$ littermates were generated by crossing $Fmr1^{+/-}$ females with $Fmr1^{+/y}$ male mice from the same production, and the resulting progeny used for our experiments was either $Fmr1^{+/y}$ (wild type) or $Fmr1^{-/y}$ (KO). Mice were maintained in collective cages following weaning (3–5 litter males per cage). Cages were balanced for genotype and supplemented with minimal enrichment (cotton nestlets). Number of mice are given in the

figure captions. The genotype of experimental animals was re-confirmed post hoc by tail-PCR.

**Surgery.** Mice (P26–42) were anaesthetized with a mixture of ketamine (100 mg kg$^{-1}$) and xylazine (10 mg kg$^{-1}$) injected intraperitoneally and supplemented as necessary throughout the procedure. Proper depth of anesthesia was monitored by testing the absence of a foot-pinch reflex and whisker movement. Mice were head-fixed using non-puncture ear-bars and a nose-clamp (SR-6M, Narishige). Body temperature was maintained at 37 °C. Prior to making an incision on the skin to expose the skull, 0.1 ml of a 1:4 Lidocaine to saline solution was administered subcutaneously and waited for 2–5 min to induce local analgesia. Following a careful removal of the scalp, and the remaining tissue on the skull, a small craniotomy was made above the S1 hindpaw region (1 mm posterior and 1.5 mm lateral from Bregma, confirmed with intrinsic imaging coupled with hindpaw stimulation) using a dental drill (World Precision Instruments).

**In vivo whole-cell patch-clamp recordings.** Blind, in vivo whole-cell recordings were performed from layer 2/3 pyramidal neurons of the hindpaw region of S1 in anesthetized mice, as described previously[35,45]. Neurons were identified by their electrophysiological properties, and in some cases by their post hoc morphology. Depth of neurons was on average 263 μm from pia, ranging from 175 μm to 374 μm. There was no genotype difference in the depth of recording (WT = 261.69 ± 34.91 μm; $Fmr1^{-/y}$ = 259.72 ± 49.12 μm; $P > 0.05$, unpaired Student $t$ test). Data were acquired at 20 kHz sampling rate and low-pass filtered at 3 kHz using Dagan BVC-700A amplifier (Dagan, Minneapolis, USA), Digidata 1320 A and Clampex 10.4 software (Axon Instruments). Recording pipettes with an open-tip resistance of 4–6 MΩ were pulled from borosilicate glass using a PC-10 puller (Narishige) and filled with intracellular solution containing (in mM): 130 K-methanesulphonate, 10 HEPES, 7 KCl, 0.05 EGTA, 2 Na$_2$ATP, 2 MgATP, 0.5 Na$_2$GTP (all products from Sigma Aldrich); pH 7.28 (adjusted with KOH); osmolarity was 280 295 osm. In a subset of experiments, biocytin (1.5–2.5 mg/ml) was added to the recording solution for post hoc neuronal identification and anatomical comparison. The intracellular solution was filtered using a 0.22-μm pore-size centrifuge filter (Costar Spin-X). Cells were excluded from the analysis if the pipette access resistance exceeded 50 MΩ or the neuron was depolarized more than −50 mV.

**Hindpaw (HP) and forepaw (FP) stimulation.** Sensory responses to tactile paw stimulus were evoked by applying squared current pulses (2 ms duration, 100 V, 30 mA) to the paws via conductive adhesive strips (-1 cm$^2$) placed on top of, and underneath the HP or FP, as described previously[35,45]. These conductive strips covered the entire paw. Following the establishment of a somatic whole-cell recording configuration, the contralateral HP or FP was stimulated 40 times at an interval of <0.3 Hz.

**Neocortical application of the specific BKCa channel agonist, BMS191011.** To pharmacologically target BK$_{Ca}$ channels, we used the specific channel agonist, BMS191011 (3-[(5-chloro-2-hydroxyphenyl) methyl]−5-[4-(trifluoromethyl)phenyl]−1,3,4-oxadiazol-2(3$H$)-one, 100 μM; Tocris). A stock solution with a concentration of 50 mM BMS191011 was prepared in DMSO and stored at −20 °C. For direct neocortical application, the drug was diluted to a final concentration of 100 μM in PBS (final concentration of DMSO in PBS: 0.2%). Cortical application of BMS191011 (-1 ml) was performed at least 30 min prior to the whole-cell patch-clamp experiments. Drug allocation was semi-randomized and balanced for cage composition.

**Acoustic startle test.** The data for the whole-body startle responses to mild auditory stimuli was taken from ref. 35 and re-analyzed for trial-by-trial and inter-individual variability. Briefly, mice ($Fmr1^{-/y}$ and WT littermates, 9–16 weeks of age) were placed in the recording chamber of a startle response box (SR-LAB, San Diego Instruments) and presented with a continuous background white noise of 65 dB. After a 5 min of habituation period, mice were exposed to 20-ms pulses of white sound of varying intensity ranging from +6 to +24 dB over background levels (equivalent to 71, 77, 83 and 89 dB). Each intensity was presented 8 times in a randomized order with variable inter-pulse intervals ranging from 10 s to 20 s. For pharmacological rescue experiments, mice were treated with either BMS204352 (Tocris) or vehicle (standard saline solution (0.9 M NaCl) supplemented with 1.25% DMSO and 1.25% Tween 80). BMS204352 (2 mg/kg) and vehicle were delivered by i.p. injection, 30 min prior to behavioral testing.

## Data analysis

**Neuronal morphology.** Following biocytin (1.5–2.5 mg/ml Biocytin, Sigma) filling of the neurons during recording, mice were perfused for post hoc staining[35]. Briefly, mice received a lethal dose of pentobarbital (300 mg/kg, i.p.) delivered in the presence of lurocaine (30 mg/kg; i.p.). Following respiratory arrest (and after verifying the absence of reflexes to toe/tail pinch and eye-blink) tissue was fixed by trans-cardial perfusion with 1× PBS (pH 7.4), followed by 4% paraformaldehyde in 1× PBS (pH 7.4). Brains were post-fixed for 2 h in 4% PFA and then stored in 1× PBS until slicing. Subsequently, 80-μm-thick slices were cut using a vibratome (Leica), and the slices were stored in 1× PBS prior to staining. Biocytin was revealed using streptavidin-Alexa Fluor 555 labeling (Invitrogen). Slices were mounted in Mowiol medium and neuronal morphology was reconstructed using a Neurolucida system (MBF Biosciences) equipped with a 100× oil immersion objective lens.

**Spontaneous AP firing.** Neurons that spontaneously fired at least one action potential (AP) during a 120-s-time window were considered spontaneously active, otherwise silent. The spontaneous AP rate was calculated as the number of APs elicited during this 120-s-time window. The analysis included data from both active and silent neurons. We acknowledge the limitation of the term "silent", since these neurons would likely become "active" if we would analyze spontaneous AP firing over a longer time window. As a result, many WT neurons had spontaneous AP firing values of zero and we could therefore not include this feature in our correlation matrix and the accompanying node plot for WT neurons.

**Up- and downstates.** For up- and downstates, both "active" and "silent" cells were included in this analysis. Custom-made Python scripts were used to detect all up- and downstates during a 180-s recording period, and to quantify their duration, frequency, and membrane potential at the respective states. A pre-processing step was performed when necessary to correct for linear drifts in membrane potential. Our algorithm annotated each point of the signal as either an up- or downstate with no intermediate state. A gliding threshold was calculated every second as the median of all points during both a 4-s-period before and after that point. For each point of the signal, the median of the surrounding points (during 50 ms before and after) was computed and compared to the corresponding gliding threshold. If this median was greater than this threshold, the point was considered part of an upstate and vice versa. Our analysis revealed "micro"-upstates lasting between 100 and 150 ms. Events lasting less than 100 ms were considered too short and removed from the analysis.

**Power of membrane potential oscillations.** The periodograms were obtained utilizing the Welch function of the Python open-source library, SciPy. Parameters such as a 4-s Hann sliding window, a 50% overlap, and the mean periodogram as the averaging method were used to calculate the Power-spectral density (PSD). PSD values for each delta (0.5–4 Hz), theta (4–7 Hz), alpha (8–12 Hz), beta (13–30 Hz), and

gamma (30–100 Hz) bands were computed by calculating the area under the curve of the periodogram to the respective frequency band by applying the composite Simpson rule.

**Membrane potential fluctuations/wavelet analysis.** Spontaneous resting signals were transformed using a complex Morlet wavelet with 4 Hz as mother wavelet frequency[39]. The widths used to scale the wavelets were computed using the following equation:

$$\frac{(w * Fs)}{(2 * yScale * \pi)} \qquad (1)$$

where $w$ is the mother wavelet frequency, $Fs$ the sampling rate (20 kHz), and $yScale$ is scale of the frequencies we are interested in. Absolute values were plotted in color-code with the scale ranging from 0 to 3. This maximum is a tradeoff between being able to detect the differences between genotypes and not saturating the signal.

**Intrinsic properties.** To study the intrinsic properties of the recorded neurons, we measured the membrane potential responses to 500-ms long step current injections ranging from −450 pA to 550 pA (step size: 50 pA). To determine the action potential (AP) threshold, we measured the membrane potential where the slope of its rising phase exceeded 10 mV/ms. AP half-width was determined by measuring the duration of the first AP at half-maximal amplitude (half-distance from threshold to peak) following the rheobase injection. Maximum AP frequency was calculated from the voltage trace with the largest number of APs. Calculation of AP accommodation was performed using a voltage trace encompassing 5 APs. Briefly, the spike interval (SI, in ms) between the 1st and 2nd AP (1st spike interval, SI), and the 4th and 5th AP (4th SI) were calculated, and AP accommodation was then calculated as 4th SI/ 1st SI. For analysis of the AP after-depolarization (ADP), trains of three APs at various frequencies were generated by brief somatic current injections (1 nA, 1.08 ms). Only AP trains occurring during downstates were selected for the analysis. Three to six trials were averaged, and the ADP amplitude (from baseline) was measured 5 ms after the peak of the last AP. AP half-width ratio was measured as the ratio of the third and first AP. To measure input resistance, we injected 500-ms-long hyperpolarizing (−100 pA) current pulses and measured the steady-state membrane potential deflection at 300 ms relative to baseline.

**Trial-by-trial and inter-individual variability of startle responses.** The data was taken from a previous study[35] and re-analyzed to measure trial-by-trial variability. The trial-by-trial variability was computed as the standard deviation of the startle responses for each auditory stimuli amplitude. Inter-individual variability was calculated as variance, and the difference in variance was tested using the Bartlett test.

**EPSPs and signal-to-noise (SNR) ratio.** Parameters of HP stimulus-evoked excitatory postsynaptic potentials (EPSPs) from 40 successive trials were calculated for EPSP-only neurons (neurons responding to HP stimulus exclusively in a sub-threshold manner, i.e., an EPSP or a failure) using Clampfit software (version 11.1, Molecular Devices, LLC). Briefly, the maximum EPSP amplitude was determined for each trial during a 200-ms time window following the HP stimulation. Trials with a response amplitude of less than two times the standard deviation of the baseline were considered as failures. EPSP duration was calculated by measuring the width of the response at half-maximal amplitude. Response slope was estimated as the rise slope between the 20th and 80th percentile of the EPSP amplitude relative to the baseline. Baseline membrane potential ($V_m$) variance was calculated as the standard deviation (SD) of the $V_m$ fluctuation during a 200-ms-time window just before the stimulus onset. The signal-to-noise ratio (SNR) was calculated similarly as described in ref. [6], by dividing the EPSP amplitude of each trial by the EPSP amplitude variance across all trials for each cell.

EPSP latencies were measured for the averaged response for each cell. EPSP onset latency was measured as the delay following HP stimulation where the Gaussian fit of the response's rising phase crosses the $V_m$ baseline (averaged $V_m$ potential during 200 ms before stimulus onset). Peak latency was calculated as the delay of the EPSP maximum amplitude with respect to the onset of the response.

**Evoked APs.** Neurons were included in the evoked AP analysis if HP stimuli elicited at least one AP during the 40 trials. Accordingly, these neurons were classified as AP-EPSP neurons. The quantification of evoked AP responses was adapted from refs. [35,45]. Briefly, spontaneous AP firing (pre-stimulus APs) was calculated as the number of APs elicited within a 200-ms-time window prior to HP stimulus. The evoked AP firing was quantified as the difference between the number of APs fired within a 200-ms long time window following the HP stimulation (post-stimulus APs) and the pre-stimulus AP number (evoked APs = post-stimulus APs – pre-stimulus APs). The coefficient of variation (c.v.) was calculated by dividing the standard deviation of AP firing by the mean evoked AP firing for individual trials. Mean AP number per successful trial was determined by dividing the number of APs evoked during a 40-trial session by the number of trials eliciting at least one AP. To determine AP dispersion, we measured the onset of the first AP in each trial within a 70-ms-time window following HP stimulus.

**Correlation matrix and node plot.** The correlation graphs were created with python custom-made scripts using NetworkX and Netgraph libraries. Seven categories of parameters (in WT neurons six, since spontaneous AP firing could not be included, see above) were defined: Trial-by-trial variability parameters, up-/downstate parameters, spontaneous AP firing, AP parameters, membrane potential ($V_m$) variance parameters (PSD + SD baseline $V_m$ variance), SNR, and EPSP parameters. Parameters were ordered depending on these categories, and each category is displayed in a different color in the graph. The nodes were arranged on a circular layout and the size of the nodes is proportional to their degrees−in this case, the number of statistically significant correlations. Only correlations with a $P$ value < 0.05 using the Pearson test are shown. Edge size and color depend on the correlation coefficient, larger coefficients (absolute value) have edges with greater width and darker color (blue for negative and red for positive correlations).

**Trial correlation parameters.** The time window chosen to compute $V_m$ baseline parameters (baseline $V_m$, baseline $V_m$ variance, PSD) on a trial-by-trial basis was a range of 200 ms before the onset of the HP stimulus. To estimate the influence of baseline $V_m$ variance and PSD on the strength, duration, and reliability of HP stimulus-evoked EPSPs, these parameters were normalized by the baseline $V_m$. For correlating these parameters for each trial, we used Pearson correlation tests.

**Overall experimental design and analysis.** Sample sizes were determined based on our published work[35,45]. In addition, we performed post hoc statistical tests of power. Mice of both genotypes were littermated and randomly assigned. Recordings and analysis were performed blind to the genotype.

### Statistical analysis
Values were first tested for outliers (Grubb's outlier test with alpha = 0.05). These outliers were removed from the statistical analysis and the resulting plots. Values were also tested for normality using the Shapiro−Wilk normality test. If the values were normally distributed an unpaired $t$ test was used to compare the two groups. For non-normally distributed parameters we used Mann−Whitney's $U$ test. A mixed ANOVA model was used for repeated measurements. As we combined silent neurons (no firing in 2-min time window) and active neurons for the calculation of spontaneous properties, we performed a two-sided

nonparametric permutation test to calculate the *P* value. Box plots indicate the median value (middle line), the mean (green line), as well as the 25th and 75th percentiles (box). The lower whisker will extend to the first datum greater than Q1 − 1.5*IQR where IQR is the interquartile range (Q3–Q1). Similarly, the upper whisker will extend to the last datum less than Q3 + 1.5*IQR (matplotlib box plot function default parameters). Correlation matrices were made with R-Pearson tests, resulting in a coefficient of correlation and an associated *P* value. Trial-by-trial variability was calculated as standard deviation of the parameter values across all trials for each cell. The F test of equality of variances or Bartlett test were used to explore the difference in variance between genotypes at the cell-population level (trial-wise average) for normally distributed data. For non-normally distributed data the Levene test was used with the mean as center parameter. Density plots (Rugg plots) were made with a Gaussian kernel density estimation using the function scipy.kde.gaussian_kde from the python library scipy. *P* values < 0.05 were considered significant (*$P < 0.05$, **$P < 0.01$, ***$P < 0.001$).

### Reporting summary

Further information on research design is available in the Nature Portfolio Reporting Summary linked to this article.

## Data availability

The electrophysiology data generated in this study have been deposited in the figshare database under accession code https://doi.org/10.6084/m9.figshare.24459607.v1. Source data are provided with this paper.

## Code availability

Custom-made Python codes used in this study can be found on the following GitHub repository: https://github.com/ToGauvrit/ElectroPhyAnalysis.

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

## Acknowledgements

We would like to thank Drs. M. Brecht, Troy Margrie, and Jérôme Epsztein for help with setting up the in vivo patch-clamp techniques in our laboratory. We thank Dr. Katy Le Corf, Dr. Rémi Proville, and Mrs. Sreedevi Madhusudhanan for assisting with analyses and immunohistochemistry. We thank Drs. Ede Rancz, Cyril Herry, Yves LeFeuvre, and Ourania Semelidou for feedback on the manuscript. We thank the animal housing and genotyping facilities of INSERM U1215 Neurocenter Magendie. This project was funded by ENC network-European Erasmus Mundus Joint PhD Fellowship (A.A.B.), Fondation pour la Recherche Médicale PhD extension grant FDT20170437003 (A.A.B.), Fondation pour la Recherche Médicale postdoctoral grant SPF20130526794, ING20140129376 (G.B.), INSERM (A.F.), Marcel Dassault-Fondation FondaMental Award 2019 (A.F.), Simons Foundation Autism Research Initiative (A.F.), and Fondation de France (A.F.).

## Author contributions

A.F. conceived the project. A.F., A.A.B., and G.B. designed the experiments. A.A.B., G.B., and and Y.V. performed the experiments. A.A.B., T.G., Y.V., and G.B. analyzed the data. T.G. developed Python computational codes for some of the analyses. M.G. contributed to the interpretation of the data and logistic support of the study. A.F., A.A.B., and T.G. prepared the figures. A.F., M.G., and A.A.B. wrote the manuscript, and T.G. and Y.V. provided feedback on the manuscript.

## Competing interests

The authors declare no competing interests.

## Inclusion and diversity statement

We support inclusive, diverse, and equitable conduct of research. We tried to use inclusive language as much as possible.
