## [Peer Review File · Nature Communications]

Endogenous noise of neocortical neurons correlates with atypical sensory response variability in the Frm1-/- mouse model of autismREVIEWER COMMENTS

Reviewer #1 (Remarks to the Author):

The authors examine the variability in sensory responses and the underlying mechanisms. The authors provide a strong argument for how their result relate to variability in sensory perception in Autism. It is a very creative study that used the powerful method of whole-cell recording to be able to understand mechanisms. They perform experiments in the primary somatosensory cortex and use paw stimulation to induce sensory responses.

It is also very complex, since the authors argue for multiple factors contributing to increased noise in the Fmr1 KO that results in sensory response variability. For the most part each of these factors is analyzed appropriately and thoroughly. But it was difficult to get a clear bottom line message. In my opinion, the strongest, most simple, novel links in the study are (related to changes in Fmr1 KO): spontaneous firing of L2/3 pyramidal neurons increased -> Vm variability is increased -> increased variability of sensory-driven EPSPs -> increased variability in latency of sensory-driven action potentials.

Initially, the changes in UP states seemed like the most clear mediator of the baseline Vm variability and the sensory response variability. But then I had 2 concerns the lessened my enthusiasm for all the text and figures devoted to it. I think it confused the message. See below.

I also disagreed with highlighting the significance of driving force being altered when in down versus up states. See comments below.

I thought the effects described for the BKca channel were described well and was totally within reason by determining some phenotypes were rescued and others not. It is a positive testimony to the experimental approach utilized that such a result was obtained and still was useful.

Major:

1) I had 2 concerns regarding UP states. First, in quiet wakefulness, is UP state frequency and duration comparable to what is observed in this study? If not, this would decrease the relevance of UP states contributing to sensory response variability. Second, while there were $\leq 20\%$ changes in duration and frequency, they were off-setting, and it seems no change in the proportion of time in the UP state. Specifically, if there is no change in the proportion of total time in UP states in the KO, variability due to

UP states would not be different. Perhaps aspects of UP state changes are correlated with response variability, but I don't see a reasonable argument for being a mechanism causing response variability.

And if the UP states don't seem like a reasonable candidate, I suppose there is some other source for variable baseline Vm perhaps stemming from the 3-100 Hz PSD (1-2 Hz would include UP state signal)? But all this suggests not as much figure and results text should be devoted to UP states since there does not seem to much there.

What is the impact of UP states that the authors are arguing? Frequency is increase by ~20% and duration decreased by <20%. Overall, the relative time in UP states would appear unchanged, and hence, the variability in EPSPs would not be expect to be significantly affected. Now if the relative time is changed, this would more greatly impact EPSP variability.

2) Line 92 and 507. Is the influence of UP states on sensory responses really due to driving force changes? I assume the authors mean the driving force at postsynaptic glutamate receptors? The UP states in this manuscript are only 5 mV in size. This would not represent much of change in driving force for excitatory transmission measured around -60 mV (0 mV reversal potential for glutamate receptors). Therefore, the change in driving force is <10% and would not significantly affect EPSP amplitude. To minimize this even more, the change in UP state amplitude in the KO is only 2.5 mV. This would very, very little impact on driving force. On the contrary, impedance might increase and have an influence?

3) Line 199. "The greater dynamicity of....resulted in a broader distribution...". I don't think this is true. The bimodal hump in the Fmr1 KO UP state region in Fig. 2o does not necessarily mean more transitions. Strictly speaking, if more transitions, there should be a relatively flat increased plateau between down and up values in the distribution. Not a bimodal distribution on the UP state. So this argument does not seem compelling to me. And what is the line drawn in Fig 2O density distribution for the KO data?

4) 2F needs better explanation in caption. It was confusing and unclear. Please indicate what exactly what the values are in the table. R2? And is this analysis for WT and Fmr1-/- combined?

5) BKca results section and figure:

For results section describing BKca agonist experiments, it should be clearly stated that the experiments were limited to Fmr1 KO mice, and that comparison data to WT and untreated Fmr1 KO mice are from previously described experiments. (If that is the case)

What was the effect on the UP state frequency decrease the Fmr1 KO?

Line 370. BK agonist results are a little confusing in presentation in fig. 5. C-F and M-Q data are normalized to Fmr1 KO data. So only 2 groups are needed for presentation. In G-L, they are not. It took me 5-10 minutes to realize this. Therefore, the Fmr1 KO data should be included in those subpanels.

Minor:

Discussion seemed a bit long.

Line 90. Is S1 the first table in the excel file? I did not know where to find this.

Line 182 – “..PSD value over this frequency range...” is this 1-100 Hz? Perhaps put this in parentheses immediately after the statement.

Line 253. Might state that this analysis is different from the TBT analysis performed earlier. Just to help the reader.... But that is up to the authors.

Reviewer #2 (Remarks to the Author):

Review of Bashkaran et al, 2023

In their manuscript entitled “Endogenous noise of neocortical neurons drives atypical sensory response variability in autism,” the authors present a dataset using in vivo patching of L2/3 neurons in somatosensory cortex of the anesthetized Fmr1 KO mouse model of fragile X syndrome (FXS). In the KO exaggerated fluctuations in the membrane potential (noise) and membrane hyperexcitability influence spontaneous and sensory evoked action potential activity, the power of oscillations in the gamma range, the frequency and duration of UP-states, and the tuning of receptive fields. These features would be predicted to alter tactile sensitivity of the mice (temporal and spatial fidelity, and detection thresholds), which unfortunately were not tested directly in these mice by behavioral methods. However, the authors do show that some sources of noise (broadened action potential width and spontaneous activity) can be dampened by activating potassium channels pharmacologically, whilst most are not (fluctuations in Vm, EPSP amplitude, UP states, and oscillations) and likely arise from altered ongoing synaptic activity in the cortex. Overall, this is a technically sophisticated electrophysiological analysis of variability in responses of layer 2/3 neurons in mouse primary somatosensory cortex in the Fmr1 KO mouse. This is a valuable contribution to the understanding of FXS.

Comments:

1. A large premise of this paper is to relate the variability seen between neurons to inter-individual variability in sensory processing in autism. The study of inter-trial variability within an animal is valuable and important, but I am less convinced by the argument about inter-individual variability. Autism is highly heterogeneous in large part to inter-individual differences in etiology (genetic and otherwise). Even within human fragile X, the basis for most inter-individual variation is genetic mosaicism. The male *Fmr1* KO mouse is not an ideal model of FXS in the sense that it is genetically homogeneous. In the current work, it is not clear that there is any inter-individual variation that cannot be accounted for fully by inter-trial variability within each mouse. Indeed, most datasets in Figures 1, 2, and 3 include multiple cells from the same animal. In some cases, it seems like even three cells were captured per animal. This confounds variability between the cells in a single animal with variability across animals. I am not convinced this paper sheds any light on inter-individual variability, and unless the authors have a better argument for why it does, they should temper their claims and restrict them to trial-by-trial variability. Overall—and this is a major criticism of the paper as written—they are over-selling their findings. If they wish to speculate beyond what the data allow, it is more appropriate for a review or opinion article.

2. To actually make the case for relevance to inter-individual variation, the authors should show that there is no significant variability between multiple cells from the same animal and that variability therefore arises from inter-individual differences. Doing this would clarify several figures. For example, in Figure 1 there seems to be almost a bimodal distribution in Fragile X cells for both EPSP and AP responses. The same bimodal distribution is also present in Figure 2C and E. Are the cells represented from one mode of the distribution primarily from one sub-population of Fragile X animals? The authors need to address the meaning of this bi-modality in one way or another. This issue really comes to a head in Figure 3I-M, where the authors are specifically comparing inter-neuronal variability and relating it to variability between humans. This is a bridge too far.

3. The authors report an elevation in gamma oscillations of membrane potential. Differences in EEG gamma power have been demonstrated previously in the auditory cortex of both mice and humans, and are believed to represent a useful biomarker in FXS. Although the finding here in somatosensory cortex is not entirely novel, to my knowledge no previous study has shown this at the level of single neurons. However, it is important to note that the anesthetic used in these experiments, ketamine, also increases cortical gamma. The authors need to acknowledge this possibility of a drug by genotype. interaction and cite the literature showing the increase in gamma and change in PV+ cell activity in Fragile X and in WT under the influence of ketamine.

4. The experiment in Figure 5 needs major revisions. First, the way they present their data does not pass statistical scrutiny. A more appropriate way to analyze this data would be to use a one-way ANOVA comparing the wild-type, knock-out, and knock-out + drug groups, not to normalize to the knock-out group. It seems that presenting the data this way will result in no significant difference between the wild-type and knock-out groups, so this study fails to reproduce their original findings and is at best

underpowered. Furthermore, the experiment is missing critical control groups. Variability could be increased by washing solution onto the brain, so a saline+knock-out group is needed, as well as ideally a wild-type+drug control. The minimum recommendation is to fix the statistical analysis and increase the sample size, the full recommendation is to also include control groups.

5. The title, introduction, and conclusion of this paper are set up to discuss variability across autism, but the authors only have data related to FXS, not idiopathic autism. Again, they are overselling the relevance of their study. The recommendation is that these sections should be re-written to focus on variability of sensory responses specifically in the FX clinical population, as this is only what their data can speak to. They should cite the relevant literature showing which sensory phenotypes are variable and which are less heterogenous in FXS. If they want to broaden the claims, they should compare the current results with other mouse models of genetic disorders that lead to autism, similar to the approach taken by Antoine et al (2019) in which 4 autism mouse models were compared.

6. Along the same lines, the authors make some relatively large leaps between the data they collect and grand conclusions about sensory responsiveness, network properties, or biomarkers. An EPSP is not a full readout of sensory processing. Intrinsic variability of neurons only correlates with variance in sensory responsiveness, it doesn't drive it. Reading out the power spectrum from a single cell's membrane potential oscillations is not representative of the oscillations in the network, unless the authors can cite a source showing the same changes and variability happen at the network level. Building this kind of bridge would help substantiate their claims that the trial by trial variability they see in cells in anesthetized animals could relate to an EEG biomarker of trial by trial variability in human patients. The recommendation is that the authors should substantially revise their language to only make claims and connections that are fully supported by the data.

7. While Figure 1T is not reported as significant, it looks like a bimodal distribution and a strong trend. This trend and distribution are worthy of comment. Please report the statistic.

8. The information presented in figure 2F was difficult to interpret with this method of presentation. Perhaps the authors could consider using a graph? Additionally, the y-axis of Figure 2J makes the data very difficult to see. The recommendation is revise to a smaller axis.

9. The data in Figure 4 is interesting but overwhelming in its presentation. Perhaps the authors could break the plot down into subplots showing the four points made in the text? Or perhaps just show significant correlations, or positive or negative correlations separately. Something to help the reader ascertain a bit more detail than simply that there are more correlations in the Fragile X neurons.

Reviewer #3 (Remarks to the Author):

This study proposes that excessive variability in sensory responses in autism originates from endogenous neural noise (i.e., variability in spontaneous network activity) in sensory cortex. They test this in *Fmr1*^{-/y} mice, by measuring many aspects of single-neuron spontaneous activity, intrinsic excitability, and sensory responses using *in vivo* whole-cell recordings in the hindpaw region of S1. This is high quality data, using a challenging method.

They find many changes *Fmr1*^{-/y} mice, including in spontaneous activity (increased Vm variability, increased spontaneous spike rate, altered up-down state dynamics, increased Vm difference between up and down states, increased power in many frequency bands), in intrinsic excitability (increased F-I curves and broader spikes), in sensory-evoked responses (increased and more variable EPSP magnitude, more variable spike latency, broader receptive fields), and more. Overall, the findings reveal a more spontaneously active, more excitable, more sensory-responsive S1, with substantially increased trial-to-trial variability. Across neurons, they find that spontaneous activity variability (captured by several measures) positively correlates with sensory response variability and sensory response magnitude (Fig. 4). Their interpretation is that more variable spontaneous activity ("endogenous noise"), coupled with increased intrinsic excitability, is what drives the variable sensory responsiveness. To test this idea, they apply a BK channel agonist that suppresses PYR excitability, and observe a reduction in stimulus-evoked responses and spontaneous firing rate.

The hypothesis that spontaneous network activity is what drives trial-by-trial sensory variability in autism makes biophysical sense, and the paper may productively shift some focus in the field to this hypothesis. However, the paper has multiple weaknesses that undercut its conclusions. Several findings are over-interpreted, for example the correlations between highly related measures of spontaneous activity (Points 1-2), and the BK agonist results (Point 3). Another logical weak point is the assumption that trial-to-trial and neuron-to-neuron variability, as observed in *Fmr1* mice, can explain inter-individual variability of sensory phenotypes in autism (Point 4).

Major

1. Much interpretation relies on correlations between variables that are not mathematically independent (e.g. Vm variability, which will have algebraic contributions from up-down Vm difference, up-down state durations, and the frequency of up-down transitions; and which will also be measured as oscillatory power in various frequency bands). Thus many of the correlations in Fig. 4 are inescapable, because these are non-independent variables. The text cites each of these correlations as an independent finding, e.g. on line 536: "Our findings suggest that key parameters of endogenous neural noise, especially Vm variance, up-down-state Vm, and oscillatory power are altered..." These are not independent pieces of evidence.

2. Throughout most of the paper, the authors are careful to not mistake correlation for causation. But this breaks down in the discussion, e.g. "an increased power in network oscillations and intrinsic

excitability strongly contribute to an increase in Vm variance" (line 440), and, importantly, in the title "Endogenous noise of neocortical neurons drives atypical sensory response variability in autism". The BK agonist experiment does not prove causality (see Point 3). Fmr1 translationally regulates dozens of downstream genes in many or all neuron types. Thus correlations could arise from the multiple cellular consequences of Fmr1 knockout, rather than propagation of a single activity feature.

3. Interpretation of the BK channel agonist (BMS) result is problematic. The discussion claims: "This finding of a localized pharmacological intervention effect supports the idea that some of the neurobiological alterations of atypical sensory information processing in autism arise at the level of S1 and are related to BK channel dysfunction". This is not true. Just because BMS reduces sensory responses and spontaneous activity does not mean that these phenotypes were caused by BK channel dysfunction, or even originate in these cells. A different pathophysiological mechanism could increase network activity, and could still be qualitatively reversed by BMS application, or any other manipulation that tamps down network excitability.

4. The paper argues that trial-to-trial variability within a neuron is related to variability between neurons, and that these mechanisms somehow cause variability across individuals--hence the broad motivation from individual variability in sensory phenotypes across people with autism (in Abstract, Introduction, Discussion). The study only measures variability at the neuronal level, not variability across individuals. Whether inter-neuronal variability drives inter-individual variability is entirely unknown, and may not be true, given that brains average across millions of neurons. Inter-individual variability could arise from entirely different processes, involving genetic, environmental or experience factors that differ between people.

Minor

5. Variations in up-down state structure can also result from different levels of anesthesia. How was it ensured that anesthetic state was the same in Fmr1 vs WT mice?

6. The intrinsic excitability experiments (Fig. 3D-H) were presumably performed with network activity intact (i.e., no synaptic blockers). Thus higher spontaneous network activity could have contributed to the increased F-I curves in Fmr1 (Fig. 3E).

7. Figure 6 does not seem useful. Box 1 lists the only two possible sources of variability in neurons. Box 2 is a fully connected graph suggesting that all noise sources drive each other. Box 3 lists both increased variability and reduced reliability as separate consequences. But what is reduced reliability except increased variability? Most of the legend is a lengthy claim about the potential implications of the model for clinical and preclinical research, which does not seem appropriate for a figure legend.

REVIEWER COMMENTS

Reviewer #1 (Remarks to the Author):

Overall assessment:

The authors examine the variability in sensory responses and the underlying mechanisms. The authors provide a strong argument for how their results relate to variability in sensory perception in Autism. It is a very creative study that used the powerful method of whole-cell recording to be able to understand mechanisms. They perform experiments in the primary somatosensory cortex and use paw stimulation to induce sensory responses.

Response:

We appreciate this overall very positive assessment of our study.

Overall assessment:

It is also very complex, since the authors argue for multiple factors contributing to increased noise in the *Fmr1* KO that results in sensory response variability. For the most part each of these factors is analyzed appropriately and thoroughly. But it was difficult to get a clear bottom line message. In my opinion, the strongest, most simple, novel links in the study are (related to changes in *Fmr1* KO): spontaneous firing of L2/3 pyramidal neurons increased -> Vm variability is increased -> increased variability of sensory-driven EPSPs -> increased variability in latency of sensory-driven action potentials.

Initially, the changes in UP states seemed like the most clear mediator of the baseline Vm variability and the sensory response variability. But then I had 2 concerns the lessened my enthusiasm for all the text and figures devoted to it. I think it confused the message. See below.

I also disagreed with highlighting the significance of driving force being altered when in down versus up states. See comments below.

I thought the effects described for the BKca channel were described well and was totally within reason by determining some phenotypes were rescued and others not. It is a positive testimony to the experimental approach utilized that such a result was obtained and still was useful.

Response:

We are grateful for this general assessment as well as constructive criticism and have addressed each point in detail. We also agree with the proposed link, but would like to suggest the following modifications: (1) In addition to spontaneous AP firing, the increased power of synaptic input oscillations in the <100 Hz range (and also aperiodic components) will contribute to the Vm variance. (2) We have now added additional figure graphs highlighting the relationship between EPSP amplitude and Vm (Fig. 2L, S), as well as EPSP amplitude and Vm variance (Fig. 2F), at the time of stimulus arrival. These panels illustrate that there is a greater range of EPSP amplitudes in *Fmr1*^{-/-} neurons in accordance with the significant increase in up-down-state Vm difference for these neurons. In addition, the greater up-down-state dynamicity in these neurons, including brief up-states and intermediate up-state Vm values, will further increase the variability of the EPSP amplitude on a trial-by-trial basis. This increased dynamicity will also impact the temporal processing of tactile information, thus resulting in a greater variability of the onset of sensory-driven action potentials.

Major Comments:

1) I had 2 concerns regarding UP states. First, in quiet wakefulness, is UP state frequency and duration comparable to what is observed in this study? If not, this would decrease the relevance of UP states contributing to sensory response variability. Second, while there were $\leq 20\%$ changes in duration and frequency, they were off-setting, and it seems no change in the proportion of time in the UP state. Specifically, if there is no change in the proportion of total time in UP states in the KO, variability due to UP states would not be different. Perhaps aspects of UP state changes are correlated with response variability, but I don't see a reasonable argument for being a mechanism causing response variability.

And if the UP states don't seem like a reasonable candidate, I suppose there is some other source for variable baseline Vm perhaps stemming from the 3-100 Hz PSD (1-2 Hz would include UP state signal)? But all this suggests not as much figure and results text should be devoted to UP states since there does not seem to much there.

What is the impact of UP states that the authors are arguing? Frequency is increase by $\sim 20\%$ and duration decreased by $< 20\%$. Overall, the relative time in UP states would appear unchanged, and hence, the variability in EPSPs would not be expect to be significantly affected. Now if the relative time is changed, this would more greatly impact EPSP variability.

Response:

There are only few *in vivo* studies that have quantified up-states during quiet wakefulness at the single neuron level in the neocortex. The study by Gonçalves et al. (Gonçalves JT, Anstey JE, Golshani P, Portera-Cailliau C. Circuit level defects in the developing neocortex of Fragile X mice. *Nat Neurosci.* 2013 Jul;16(7):903-9) serves as our reference point for comparison. However, direct comparison with this and other studies is difficult due to the different analysis methods used. In particular, in our study we were intrigued by the frequent presence of brief up-states ($< 150\text{ms}$) in *Fmr1*^{-y} neurons when compared to WT neurons. This prompted us to tailor our analysis to specifically detect and quantify the fine-details of these fractionated up-states, in contrast to algorithms used in other studies, which will not detect up-states shorter than 1 second. Thus, on the basis of the available literature we cannot make this comparison.

In this context, it might be interesting to note one study reported the presence of brief up-states in mice lacking NMDA receptors in parvalbumin-positive interneurons (which have been implicated in up-down-state transitions) (Guyon et al., 2021, *The Journal of Neuroscience*, 41(13):2944–2963).

Although we do not observe a genotype difference in the proportion of time spent in the up-state, we found that the brief up-states often do not return to the main down-state but rather reach an intermediate level, resulting in a broader range of up-state Vm values (Fig. 2R). Given the strong reliance of EPSP amplitude on the Vm, this broader range of up-states together with the overall increased up-down-state Vm difference likely contributes to a broader and more variable range of sensory-driven EPSP amplitudes in *Fmr1*^{-y} neurons (see also above).

We agree with the reviewer that the increase in 3-100 Hz PSD is a main source for increased Vm variance in *Fmr1*^{-y} neurons. In addition to the PSD, aperiodic components will also contribute to Vm variance. Vm variance during up-states, in turn, can induce state transitions to a down-state (Jercog, D. et al., 2017, *eLife*), and therefore be a source of the enhanced up-down-state transition dynamics in these neurons.

2) Line 92 and 507. Is the influence of UP states on sensory responses really due to driving force changes? I assume the authors mean the driving force at postsynaptic glutamate

receptors? The UP states in this manuscript are only 5 mV in size. This would not represent much of change in driving force for excitatory transmission measured around -60 mV (0 mV reversal potential for glutamate receptors). Therefore, the change in driving force is <10% and would not significantly affect EPSP amplitude. To minimize this even more, the change in UP state amplitude in the KO is only 2.5 mV. This would have very, very little impact on driving force. On the contrary, impedance might increase and have an influence?

Response:

We thank the Reviewer for this comment and agree that other factors might have an influence as well. We have now included new figure panels to strengthen our findings. Indeed, we refer to the driving force for ion flux through the postsynaptic glutamate receptors. It is true that the absolute up-down-state Vm difference between genotypes in our data is not very large, albeit statistically significant, suggesting that it may perhaps not strongly affect driving force. However, our data shows a strong relationship between EPSP amplitude and Vm (new Fig. 2L), and a positive correlation between the range of EPSP amplitudes and the range of Vm values at the time of arrival of the sensory signal (new Fig. 2S). Fig. 2S shows that the neurons with the largest baseline Vm difference are also those with the largest EPSP amplitude range – and those are *Fmr1*^{-/-} neurons. It is conceivable that the driving force difference is more pronounced in the dendrites receiving the synaptic input. It is also possible that impedance differences between up- and down-states strongly influence synaptic responses. We have now modified the text accordingly to include impedance as a contributing factor.

3) Line 199. “The greater dynamicity of....resulted in a broader distribution...”. I don’t think this is true. The bimodal hump in the *Fmr1* KO UP state region in Fig. 2o does not necessarily mean more transitions. Strictly speaking, if more transitions, there should be a relatively flat increased plateau between down and up values in the distribution. Not a bimodal distribution on the UP state. So this argument does not seem compelling to me. And what is the line drawn in Fig 2O density distribution for the KO data?

Response: We apologize for not being clear on this point. The panel O of Figure 2 (new Fig. 2R) is an illustration of what the boxplot in panel P (new Fig. 2Q) is showing, namely that for the *Fmr1*^{-/-} group the difference between up- and downstate is increased and more variable compared to the WT group. The values were calculated by normalizing each value of upstate Vm by the average downstate Vm for each neuron. The top part of the plot shows that the distribution is flatter for the *Fmr1*^{-/-} group. The bottom part of the plot visualizes the range of normalized upstate Vm values for each neuron, demonstrating a greater range of normalized up-state Vm values for the *Fmr1*^{-/-} group. We agree with the Reviewer that we cannot link the upstate frequency with this broader range of upstates Vm values. We therefore changed the sentence accordingly: “We observed a greater range of upstates Vm values in *Fmr1*^{-/-} neurons (new Fig. 2R).” The lines in the density distribution plot mark the distribution of both WT and *Fmr1*^{-/-} values and indicate the overlap between both distributions.

4) 2F needs better explanation in caption. It was confusing and unclear. Please indicate what exactly what the values are in the table. R2? And is this analysis for WT and *Fmr1*^{-/-} combined?

Response:

We apologize for the confusion. The values in the table are mean correlation coefficients for WT and *Fmr1*^{-/-} neurons combined, since we did not observe a genotype difference for these values. The coefficients show a negative correlation between the baseline Vm and the EPSP amplitude and halfwidth. Thus, the trials with the most hyperpolarized baseline Vm displayed the largest EPSP amplitude/halfwidth. Consequently, a larger range of EPSP amplitudes could

be explained by a larger range of baseline Vm values in *Fmr1*^{-/-} neurons. In addition, we show that the EPSP amplitude and EPSP halfwidth positively correlate with the baseline Vm variance (normalised by the Vm). Thus, trials with larger baseline Vm variance also exhibit larger EPSP amplitudes/halfwidths. Altogether, our data suggest that the up-down-state Vm dynamics and the baseline Vm variance strongly contribute to the increased trial-by-trial variability of *Fmr1*^{-/-} neurons.

5) BKCa results section and figure:

For results section describing BKCa agonist experiments, it should be clearly stated that the experiments were limited to *Fmr1* KO mice, and that comparison data to WT and untreated *Fmr1* KO mice are from previously described experiments. (If that is the case)

What was the effect on the UP state frequency decrease the *Fmr1* KO?

Line 370. BK agonist results are a little confusing in presentation in fig. 5. C-F and M-Q data are normalized to *Fmr1* KO data. So only 2 groups are needed for presentation. In G-L, they are not. It took me 5-10 minutes to realize this. Therefore, the *Fmr1* KO data should be included in those subpanels.

Response:

We apologize for not being clear enough about the presentation of this data. Indeed, the data for the WT and *Fmr1*^{-/-} groups were the same as used in the previous figures of this study, while the *Fmr1*^{-/-} + BMS group was a new data set. We have added a statement that WT and *Fmr1*^{-/-} data are taken from previously described experiments.

BMS application did not rescue the elevated up-state frequency of *Fmr1*^{-/-} neurons, but rather increased it (Table S4).

We thank the Reviewer for this suggestion. We have now restructured our data. Instead of presenting the data as values that were normalized to the *Fmr1*^{-/-} group to statistically test for rescue, we now plotted all three groups (new Fig. 5). For statistical comparison between these three groups, we used ANOVA test as well as Bonferroni or Dunn post-hocs tests.

Minor:

Comment:

Discussion seemed a bit long.

Response:

We thank the Reviewer for pointing this out, and have made an effort of shortening the discussion.

Comment:

Line 90. Is S1 the first table in the excel file? I did not know where to find this.

Response:

Originally, we combined two excel sheets into a single table (Supplementary table S1), but for clarity we now decided to split it into two separate tables (S1 and S2). Table S1 contains all the values analysed from our single neuron recordings, while table S2 contains the variance (cell-by-cell variance). We have also added table S3 (mouse-by-mouse variance) and table S4 (multiple comparison for BMS treatment).

Comment:

Line 182 – “..PSD value over this frequency range...” is this 1-100 Hz? Perhaps put this in parentheses immediately after the statement.

Response:

We added the frequency range in parenthesis.

Comment:

Line 253. Might state that this analysis is different from the TBT analysis performed earlier. Just to help the reader.... But that is up to the authors.

Response: We would prefer to keep the sub-header as it is if the Reviewer agrees.

Reviewer #2 (Remarks to the Author):

Overall assessment:

Review of Bashkaran et al, 2023

In their manuscript entitled “Endogenous noise of neocortical neurons drives atypical sensory response variability in autism,” the authors present a dataset using in vivo patching of L2/3 neurons in somatosensory cortex of the anesthetized Fmr1 KO mouse model of fragile X syndrome (FXS). In the KO exaggerated fluctuations in the membrane potential (noise) and membrane hyperexcitability influence spontaneous and sensory evoked action potential activity, the power of oscillations in the gamma range, the frequency and duration of UP-states, and the tuning of receptive fields. These features would be predicted to alter tactile sensitivity of the mice (temporal and spatial fidelity, and detection thresholds), which unfortunately were not tested directly in these mice by behavioral methods. However, the authors do show that some sources of noise (broadened action potential width and spontaneous activity) can be dampened by activating potassium channels pharmacologically, whilst most are not (fluctuations in Vm, EPSP amplitude, UP states, and oscillations) and likely arise from altered ongoing synaptic activity in the cortex. Overall, this is a technically sophisticated electrophysiological analysis of variability in responses of layer 2/3 neurons in mouse primary somatosensory cortex in the Fmr1 KO mouse. This is a valuable contribution to the understanding of FXS.

Response:

We are grateful to the Reviewer for this overall very positive evaluation. We also agree with the Reviewer’s view that adding behavior would lift the story. We have therefore decided to include new analysis from a sensory behavioral task (sensory stimulus evoked body startle response), evaluating trial-by-trial variability and inter-individual variability at the behavioral level (new Fig. 1U and V; new Fig. 3N). We also include here new data from another behavioral task (a modified version of the adhesive tape removal test that evaluates sensory sensitivity/reactivity; see figure below). We are open to including this data in the manuscript if the Reviewers decide it would strengthen our findings.

Figure. Tape removal test reveals larger population variance in $Fmr1^{-/-}$ mice. (A) Schematic of the position of adhesive tape strips on both hindpaws of the mouse. (B) Schematic of a mouse removing the tape on the left hindpaw. (C) No difference in the time to first contact between the WT and $Fmr1^{-/-}$ mice. (D) Greater mouse-to-mouse variance in the $Fmr1^{-/-}$ population for time to first contact with the tape ($Fmr1^{-/-}$, $n = 9$ mice; WT, $n = 11$ mice). Statistical significance was calculated using unpaired t -test (C) or Bartlett variance test (D). n.s., not significant, $*P < 0.05$, $**P < 0.01$.

(Methods for Adhesive tape removal test

We performed the adhesive tape removal test as explained in (Bouet et al., 2009). Briefly, the adhesive removal test is used to study the sensory-motor deficits related to the paw and mouth. Mice were habituated to the testing room at least 30 minutes prior to the experiment (experiment room was different from that of housing). Then the animals were habituated to the testing box (regular transparent box used to house the mice) for another 60 seconds. Strips of adhesive tape (2mm x 2mm) were applied to the plantar surface of both the right and left hindpaws of the mice with equal pressure. After the adhesive tape placement, the animals were placed in the testing box. Measurements were taken by two experimenters (blinded to genotype) using a chronometer. Sensory reactivity was determined by quantifying the time when the mouse reacted to, and made first contact with, the adhesive tape on either the left or right paw. Difference in the variance of time to first contact between genotypes was calculated by Bartlett test).

We would also like to reply to the statement about the effects of pharmacologically activating potassium channels. We have now re-plotted the data (see response to comment 5 of Reviewer 1), showing that our pharmacological approach rescues (WT vs $Fmr1^{-/-}$, n.s.) the following physiological properties: EPSP amplitude, EPSP halfwidth, EPSP onset latency, trial-by-trial variability of EPSP half-width and EPSP slope, baseline Vm variance, spontaneous AP firing, AP broadening, EPSP half-width variance, spontaneous AP variance, and trial-by-trial variability of SD startle reactivity. Some of the measures are indeed not rescued, in particular the trial-by-trial variability of the EPSP amplitude, up-down-state Vm difference, or the power of most oscillations (see Table S4).

Comments:

1. A large premise of this paper is to relate the variability seen between neurons to inter-individual variability in sensory processing in autism. The study of inter-trial variability within an animal is valuable and important, but I am less convinced by the argument about inter-individual variability. Autism is highly heterogeneous in large part to inter-individual differences in etiology (genetic and otherwise). Even within human fragile X, the basis for most inter-individual variation is genetic mosaicism. The male $Fmr1$ KO mouse is not an ideal model of FXS in the sense that it is genetically homogeneous. In the current work, it is not clear that there is any inter-individual variation that cannot be accounted for fully by inter-trial variability within each mouse. Indeed, most datasets in Figures 1, 2, and 3 include multiple cells from the same animal. In some cases, it seems like even three cells were captured per animal. This confounds

variability between the cells in a single animal with variability across animals. I am not convinced this paper sheds any light on inter-individual variability, and unless the authors have a better argument for why it does, they should temper their claims and restrict them to trial-by-trial variability. Overall—and this is a major criticism of the paper as written—they are over-selling their findings. If they wish to speculate beyond what the data allow, it is more appropriate for a review or opinion article.

Response:

We agree with the Reviewer that this is a weakness of our study and thank the Reviewer for pointing this out. We have now re-analysed our data by combining values of all cells for each mouse to be in the position to compare mouse-by-mouse variability between genotypes. While some of the measures of increased cell-by-cell variability are still maintained and expressed as increased mouse-by-mouse variability for *Fmr1*^{-y} mice (e.g., evoked APs/40 trials, delta, theta, alpha, beta and gamma frequency power, spontaneous AP firing frequency; see Tables S2 and S3), others are no longer different when compared to WT mice (e.g., AP half-width ratio, up-down-state Vm difference, EPSP half-width, EPSP onset latency, baseline Vm fluctuation, see Tables S2 and S3).

In addition, we have included new data from two behavioral tests (auditory stimulus evoked startle response and tactile stimulus dependent tape removal test). Both tests do indeed show increased inter-individual (i.e., mouse-by-mouse) variability for the *Fmr1*^{-y} group. Increased inter-individual variability in the acoustic startle test is now shown in Fig. 3N and described in the manuscript (pages 11). We are including the results from the tape removal test in this letter (see above) and are open to include it in the manuscript if the Reviewer decide it would strengthen our findings. This data shows that the time to first contact of the tape on the forepaws is more variable across the *Fmr1*^{-y} mouse population when compared to the WT group.

Nonetheless, to avoid overselling our results, we have now decided to tune down our original statement of increased inter-individual variability (see page 11).

2. To actually make the case for relevance to inter-individual variation, the authors should show that there is no significant variability between multiple cells from the same animal and that variability therefore arises from inter-individual differences. Doing this would clarify several figures. For example, in Figure 1 there seems to be almost a bimodal distribution in Fragile X cells for both EPSP and AP responses. The same bimodal distribution is also present in Figure 2C and E. Are the cells represented from one mode of the distribution primarily from one sub-population of Fragile X animals? The authors need to address the meaning of this bi-modality in one way or another. This issue really comes to a head in Figure 3I-M, where the authors are specifically comparing inter-neuronal variability and relating it to variability between humans. This is a bridge too far.

Response: We thank the Reviewer for raising an important point. Indeed, we believe that there may be subpopulations with respect to the sensory responses and had previously statistically tested this. However, our sample size is not sufficient to claim this point. We had mentioned the possibility of subgroups in the discussion (page 21) section *Dysfunctional oscillatory power reflects deficits in synaptic input patterns*. Since most of our recordings stem from a single recording per mouse, we cannot statistically compare the variability between different cells recorded from the same animal. However, we have pooled multiple recordings and analysed variability from mouse to mouse (see comment 1). We decided to change the language of the text to tone down the mouse-by-mouse variability of the neural data. Instead, we now include behavioral data attesting to increased mouse-by-mouse variability in the *Fmr1*^{-y} group for a sensory reactivity test.

3. The authors report an elevation in gamma oscillations of membrane potential. Differences in EEG gamma power have been demonstrated previously in the auditory cortex of both mice and humans, and are believed to represent a useful biomarker in FXS. Although the finding here in somatosensory cortex is not entirely novel, to my knowledge no previous study has shown this at the level of single neurons. However, it is important to note that the anesthetic used in these experiments, ketamine, also increases cortical gamma. The authors need to acknowledge this possibility of a drug by genotype. interaction and cite the literature showing the increase in gamma and change in PV+ cell activity in Fragile X and in WT under the influence of ketamine.

Response: We are grateful that the Reviewer points out the novelty of this finding at the single cell level. We found that in addition to gamma power, the power of the other major frequency bands in the 1-100 Hz range (delta, theta, beta) are also significantly increased in the *Fmr1*^{-/-} group. Regarding the effect of ketamine, it is true that ketamine has been shown to act on NMDA receptors to reduce the activity of PV+ cells, which in turn leads to an increase in the broadband gamma power (Guyon et al., 2021, The Journal of Neuroscience, 41(13):2944–2963). Similarly, dysfunction of NMDA receptors on PV+ cells can have the same effect on gamma power (Guyon et al., 2021, The Journal of Neuroscience, 41(13):2944–2963). Moreover, dysfunction of PV+ cells and an increase in gamma power has been described in FXS (Gibson et al., 2008 J Neurophysiol, 100:2615–26.1. Razak et al., 2021 Frontiers in Psychiatry, 12:1-11). This would be consistent with our findings of enhanced broadband gamma power. Given that PV+ cells might already be more dysfunctional in *Fmr1*^{-/-} mice would seem to argue against ketamine having a stronger effect on gamma power in *Fmr1*^{-/-} mice compared to WT mice and might therefore not be a major contributing factor to the enhanced gamma power. However, as far as we are aware, no study has yet specifically compared the effect of ketamine on gamma power and PV+ cell activity between *Fmr1*^{-/-} and WT mice.

4. The experiment in Figure 5 needs major revisions. First, the way they present their data does not pass statistical scrutiny. A more appropriate way to analyze this data would be to use a one-way ANOVA comparing the wild-type, knock-out, and knock-out + drug groups, not to normalize to the knock-out group. It seems that presenting the data this way will result in no significant difference between the wild-type and knock-out groups, so this study fails to reproduce their original findings and is at best underpowered. Furthermore, the experiment is missing critical control groups. Variability could be increased by washing solution onto the brain, so a saline+knock-out group is needed, as well as ideally a wild-type+drug control. The minimum recommendation is to fix the statistical analysis and increase the sample size, the full recommendation is to also include control groups.

Response:

We thank the Reviewer for this comment/suggestion. Our reasoning for presenting the data as normalized to the *Fmr1*^{-/-} neuron measures and then to compare normalized WT and *Fmr1*^{-/-}-BMS data was to test for correction of alterations. We agree with the Reviewer's suggestion to present the data as three groups and to perform an ANOVA comparing WT, KO, and KO-BMS data. This statistical analysis confirms our previous findings of correction of all but one measures. In addition, this analysis also reveals correction of additional measures related to crucial noise features (see Fig. 5).

We would also like to emphasise that using BMS was our strategy to determine if the alterations we observe in *Fmr1*^{-/-} mice can be modulated locally by targeting the S1 circuit. We used BMS as it has been shown to regulate neuronal excitability in cortical neurons in FXS mice (Zhang et al., 2014). We used local rather than a systemic BMS application to decipher which neuronal features can be locally regulated. Since local cortical BMS application is not a translatable therapeutic approach, we did not think that it was necessary to include a WT +

BMS group in our data. We therefore opted for the Reviewer's suggestion to use a different statistical analysis.

Finally, we always apply extracellular solution to the surface of the brain (in all groups), thus the genotype dependent differences in variability we observed is not due to washing solution onto the brain.

5. The title, introduction, and conclusion of this paper are set up to discuss variability across autism, but the authors only have data related to FXS, not idiopathic autism. Again, they are overselling the relevance of their study. The recommendation is that these sections should be re-written to focus on variability of sensory responses specifically in the FX clinical population, as this is only what their data can speak to. They should cite the relevant literature showing which sensory phenotypes are variable and which are less heterogenous in FXS. If they want to broaden the claims, they should compare the current results with other mouse models of genetic disorders that lead to autism, similar to the approach taken by Antoine et al (2019) in which 4 autism mouse models were compared.

Response:

We agree that FXS is not idiopathic autism and to respond to this criticism we have modified some of the terms that we used (like "autistic mice", "sensory symptoms in autism"), we have added a qualifying statement about the limitations of extending our findings to idiopathic autism, and throughout the text now refer to the *Fmr1*^{-y} model as a model for FXS and autism.

We appreciate the point that the Reviewer is making, in particular given that an estimated 70% cases of autism are idiopathic. Idiopathic autism represents a particular challenge to the preclinical field, due to unknown etiology and the lack of construct validity or problems with reproducibility of some of the proposed models of idiopathic autism. The preclinical field has thus largely focussed on genetic mouse models reflecting mutations in well-validated autism risk genes which can be easily controlled and characterized according to criteria related to construct-, face- and predictive validity. In addition, there is an interest in developing new measures that can be applied to these models to better represent the human clinical condition (e.g., Robertson and Baron-Cohen, 2017)

While we have made the changes described above, we would like to make the following points for why we believe the manuscript is still strongly relevant to autism:

- 1) The *Fmr1*^{-y} model was originally designed as a model for FXS but is now widely accepted in the preclinical field as a model for both FXS and autism (e.g., He et al., 2017, *The Journal of Neuroscience*, 37(27):6475– 6487), in particular for exploring atypical sensory experience.
- 2) The *FMR1* gene is a well-established autism risk gene (SFARI gene score=1). Recent analysis (using DSM-5 criteria and a large sample (547 children and adolescents with FXS)) demonstrate that 51% of boys with FXS also have autism as a co-occurring condition (Kaufmann et al., 2017, *Pediatrics*, 139(Suppl 3):S194-S206).
- 3) *Fmr1*^{-y} mice have strong face value as a model for both FXS and ASD (in addition to behavioural features they also replicate physiological markers in common with autism, and demonstrate highly reproducible sensory phenotypes). *Fmr1*^{-y} mice also have predictive value as they respond to non-pharmacological interventions such as environmental enrichment, which might be one of the only truly translational treatments given the absence of validated pharmacological methods for treating FXS and autism.
- 4) The FMR protein, a translational regulator, has wide-reaching effects on other genes associated with synaptic development through gene expression networks involving other ASD-associated risk genes (Darnell et al., 2011, *Cell*. 2011 Jul 22;146(2):247-61; Parikshak et al., 2013, *Cell*, 155(5):1008-21). The fact that FMRP, synaptic proteins and ASD-related genes are intertwined suggests the possibility of interwoven molecular mechanisms, leading to altered cellular and circuit function which may more generally model cell and circuit changes in autism.

Further, the notion of what constitutes a valid model for neuropsychiatric disorders is constantly evolving (e.g., see Shemesh and Chen, *Molecular Psychiatry*, 2023, 28:993–1003) and there is an impetus for improving the validity of models using not only forward translation, but also back translation approaches (i.e., comparing how the model corresponds to human clinical symptoms as discussed in Robertson and Baron-Cohen 2017). The sensory domain provides an immense potential for doing this due to the well-conserved circuitry, peripheral receptors and CNS regions involved in sensory processing. We would like to point out that our approach does exactly that, notably taking the finding of variability from human studies and seeing how the model shapes up.

Indeed, we agree that it would be ideal to compare our findings across genetic models of autism, to enable us to establish whether there are convergent features of noise that allow a broader generalization of our findings. However, the experimental approach that we have used, which allows us to access a rich set of physiological measures, is extremely painstaking and time-intensive to perform. It is our intention in the future to compare our current results to results which we hope to obtain from another genetic model, albeit using a smaller subset of parameters which we will define using the current study. However, given the time investment required for this work, we believe that this is beyond the scope of the current manuscript. Given the very detailed analysis which we provide in the current manuscript, and the novel findings of variability which have not yet been described in a preclinical model, we believe it is timely that we present this analysis to the scientific community to also inspire other research groups to explore variability as a feature of preclinical models of autism.

We appreciate the suggestion of the reviewer that we confine our discussion of what has been clinically shown in FXS patients. Unfortunately, very few studies have described neurophysiological measures of sensory processing in people with FXS and there is no literature describing variability as far as we are aware. We believe that by restricting our discussion to this very limited FXS literature, we might miss a chance to describe how the *Fmr1*^{-/-} mouse recapitulates features of variability and noise in autism.

6. Along the same lines, the authors make some relatively large leaps between the data they collect and grand conclusions about sensory responsiveness, network properties, or biomarkers. An EPSP is not a full readout of sensory processing. Intrinsic variability of neurons only correlates with variance in sensory responsiveness, it doesn't drive it. Reading out the power spectrum from a single cell's membrane potential oscillations is not representative of the oscillations in the network, unless the authors can cite a source showing the same changes and variability happen at the network level. Building this kind of bridge would help substantiate their claims that the trial by trial variability they see in cells in anesthetized animals could relate to an EEG biomarker of trial by trial variability in human patients. The recommendation is that the authors should substantially revise their language to only make claims and connections that are fully supported by the data.

Response: We agree with the reviewer that we cannot conclusively conclude from single cell data about the sensory responsiveness of the animal. Although an EPSP is not a full readout of sensory processing, it likely corresponds to early events in sensory processing (which are also suggested to be the most reliable biomarkers of altered sensory information processing in humans (Robertson and Baron-Cohen 2017)). What is striking, though, is that some of our findings replicate findings from EEG/MRI measures in humans, e.g., the trial-by-trial variability of sensory responses or the oscillation power. The strength of single-cell recordings is that we can arrive at a better mechanistic understanding of these alterations, and that we can then test pharmacological correction. To strengthen our findings, we have now included behavioral data supporting our claims of increased trial-by-trial variability (and inter-individual variability) in this model – please see response to reviewer 2, and Fig. 1U and V, and Fig. 3N. We have now modified several of our statements (see page 20 line 564-565; pages 22 and 23, section

Biomarkers).

7. While Figure 1T is not reported as significant, it looks like a bimodal distribution and a strong trend. This trend and distribution are worthy of comment. Please report the statistic.

Response: Indeed, the distribution is not normal, which is why we used a non-parametric test, the Mann-whitney test. The p value is 0.105 and the statistic is 77.0. The two medians are close to each other (WT: 1.048 ± 0.098 and KO: 1.181 ± 0.215). The KO group seems to show a bimodal distribution that might indicate the presence of two populations of responsive neurons, one group firing only a single AP in response to hindpaw stimulus, and another group firing with several APs. This increasing firing would be expected to alter the encoding of the sensory information.

8. The information presented in figure 2F was difficult to interpret with this method of presentation. Perhaps the authors could consider using a graph? Additionally, the y-axis of Figure 2J makes the data very difficult to see. The recommendation is revise to a smaller axis.

Response:

We thank the reviewer for these suggestions. In addition to Fig 2F (new Fig, 2G), we have now included three graphs (new panels Fig. 2F, L, and S) that better explain the relationship between EPSP amplitude and baseline Vm as well as baseline Vm variance. In addition, we modified Fig. 2J by using a smaller y-axis range.

9. The data in Figure 4 is interesting but overwhelming in its presentation. Perhaps the authors could break the plot down into subplots showing the four points made in the text? Or perhaps just show significant correlations, or positive or negative correlations separately. Something to help the reader ascertain a bit more detail than simply that there are more correlations in the Fragile X neurons.

Response:

We agree with the reviewer that Fig. 4 in its current form is overwhelming. Originally, we did not want to exclude any measures that showed a statistically significant correlation (either positive or negative), but we agree that visibility suffers from this approach. We therefore have now decided to limit the correlations shown in the node plots to those that we believe are central for the atypical sensory responses. In addition, we removed some of the important measures because they are mathematically linked to other measures, e.g. up-state frequency, up-state duration, down-state duration, up-state Vm, and trial-by-trial PSD. Nonetheless, all significant correlations between the measures (both positive and negative) are still listed in the correlation plots, and the main findings from this analysis are stated in the text (section *Relationship between endogenous noise and atypical sensory information processing*, pages 13 and 14).

Reviewer #3 (Remarks to the Author):

Overall assessment:

This study proposes that excessive variability in sensory responses in autism originates from endogenous neural noise (i.e., variability in spontaneous network activity) in sensory cortex. They test this in *Fmr1*^{-/-} mice, by measuring many aspects of single-neuron spontaneous activity, intrinsic excitability, and sensory responses using in vivo whole-cell recordings in the hindpaw region of S1. This is high quality data, using a challenging method. They find many changes *Fmr1*^{-/-} mice, including in spontaneous activity (increased Vm variability, increased spontaneous spike rate, altered up-down state dynamics, increased Vm

difference between up and down states, increased power in many frequency bands), in intrinsic excitability (increased F-I curves and broader spikes), in sensory-evoked responses (increased and more variable EPSP magnitude, more variable spike latency, broader receptive fields), and more. Overall, the findings reveal a more spontaneously active, more excitable, more sensory-responsive S1, with substantially increased trial-to-trial variability. Across neurons, they find that spontaneous activity variability (captured by several measures) positively correlates with sensory response variability and sensory response magnitude (Fig. 4). Their interpretation is that more variable spontaneous activity ("endogenous noise"), coupled with increased intrinsic excitability, is what drives the variable sensory responsiveness. To test this idea, they apply a BK channel agonist that suppresses PYR excitability, and observe a reduction in stimulus-evoked responses and spontaneous firing rate.

The hypothesis that spontaneous network activity is what drives trial-by-trial sensory variability in autism makes biophysical sense, and the paper may productively shift some focus in the field to this hypothesis. However, the paper has multiple weaknesses that undercut its conclusions. Several findings are over-interpreted, for example the correlations between highly related measures of spontaneous activity (Points 1-2), and the BK agonist results (Point 3). Another logical weak point is the assumption that trial-to-trial and neuron-to-neuron variability, as observed in *Fmr1* mice, can explain inter-individual variability of sensory phenotypes in autism (Point 4).

Response: We are grateful for the overall assessment of our paper and that it may have a conceptual influence on the field. We also acknowledge the mentioned weaknesses, which we address in detail.

Major

1. Much interpretation relies on correlations between variables that are not mathematically independent (e.g. Vm variability, which will have algebraic contributions from up-down Vm difference, up-down state durations, and the frequency of up-down transitions; and which will also be measured as oscillatory power in various frequency bands). Thus many of the correlations in Fig. 4 are inescapable, because these are non-independent variables. The text cites each of these correlations as an independent finding, e.g. on line 536: "Our findings suggest that key parameters of endogenous neural noise, especially Vm variance, up-down-state Vm, and oscillatory power are altered..." These are not independent pieces of evidence.

Response: We agree with the reviewer that the aforementioned parameters, in particular up-down-state Vm difference, delta power, and Vm variance are linked in some way and therefore not independent. They are part of the measures of spontaneous activity and endogenous noise. Nevertheless, there are differences between these parameters with respect to what they are measuring. For example, the value of up-down-state Vm difference is not influenced by the Vm variance. The up-down-state Vm difference contributes to the lower PSD band, but this PSD band will also measure all other periodic components in this respective frequency band, such as depolarizations that are too short (<100ms) to be considered as up-states. The baseline Vm variance is measuring all fluctuations present in the baseline: periodic (included in PSD, but also all the non-periodic components and up-down-state Vm difference. Moreover, these measures can influence each other; for example, Vm variance can generate up-down-state transitions (Jercog, D. et al., 2017, eLife).

In summary, even though these measures influence each other and are partially mathematically linked, we are mainly focusing on the link between them and the sensory evoked responses and their variability.

2. Throughout most of the paper, the authors are careful to not mistake correlation for

causation. But this breaks down in the discussion, e.g. "an increased power in network oscillations and intrinsic excitability strongly contribute to an increase in Vm variance" (line 440), and, importantly, in the title "Endogenous noise of neocortical neurons drives atypical sensory response variability in autism". The BK agonist experiment does not prove causality (see Point 3). *Fmr1* translationally regulates dozens of downstream genes in many or all neuron types. Thus correlations could arise from the multiple cellular consequences of *Fmr1* knockout, rather than propagation of a single activity feature.

Response: We agree with the reviewer and have therefore modified the respective sentences (page 1, line 25) as well as the title ("*Endogenous noise of neocortical neurons correlates with atypical sensory response variability in the *Fmr1*^{-/-} mouse model of autism*"). However, we would like to point out that an increase in the oscillation power (the power is directly linked to the amplitude of these oscillations) will directly contribute to the Vm variance.

3. Interpretation of the BK channel agonist (BMS) result is problematic. The discussion claims: "This finding of a localized pharmacological intervention effect supports the idea that some of the neurobiological alterations of atypical sensory information processing in autism arise at the level of S1 and are related to BK channel dysfunction". This is not true. Just because BMS reduces sensory responses and spontaneous activity does not mean that these phenotypes were caused by BK channel dysfunction, or even originate in these cells. A different pathophysiological mechanism could increase network activity, and could still be qualitatively reversed by BMS application, or any other manipulation that tamps down network excitability.

Response:

We agree with the reviewer that there are alternative interpretations of these results, such as alterations that arise elsewhere but are sensitive to local pharmacological modification, or changes that are sensitive to BKCa channel manipulation but based on a different mechanism. We now modified the text accordingly (page 16 (line 412), page 23 (line 663-664)). To further clarify our statement: Since BMS has poor bioavailability, its effect is relatively local, i.e. within S1. Thus, any effect we observe is due to a local (at the level of S1) effect on our measures. The effect could be either within the recorded neuron (e.g., AP widening), or on presynaptic and/or postsynaptic sites (e.g., EPSP amplitude, spontaneous AP firing). We also agree with the reviewer that the effect could either be the result of correcting a BKCa channel dysfunction – which we and others have shown to occur in *Fmr1*^{-/-} neurons – or by targeting changes due to other mechanisms within the S1 network, which are sensitive to BKCa channel modulation.

4. The paper argues that trial-to-trial variability within a neuron is related to variability between neurons, and that these mechanisms somehow cause variability across individuals--hence the broad motivation from individual variability in sensory phenotypes across people with autism (in Abstract, Introduction, Discussion). The study only measures variability at the neuronal level, not variability across individuals. Whether inter-neuronal variability drives inter-individual variability is entirely unknown, and may not be true, given that brains average across millions of neurons. Inter-individual variability could arise from entirely different processes, involving genetic, environmental or experience factors that differ between people.

Response:

We apologize for not presenting the proper analysis of our data to address the issue of inter-individual variability. We have now included additional analysis of our data and also added new experiments to strengthen our conclusion. 1. We have averaged cells for each mouse to compare values on a mouse-by-mouse basis. This shows that there is greater inter-individual variability in the *Fmr1*^{-/-} population in several parameters (Supplementary Table S3). 2. We have

performed a sensory behavioral task that validates tactile sensitivity (and dexterity). We found that there is not only an alteration in sensory reactivity (altered delay in the initialization to remove the tape on the paw), but more importantly, there is significantly greater inter-individual variability in this measure in the *Fmr1*^{-y} mouse population (See first response to reviewer 2). We have also included data from acoustic startle test see Fig. 1V that show more inter-individual variability as well (Fig. 3N; page 12)

Minor

5. Variations in up-down state structure can also result from different levels of anesthesia. How was it ensured that anesthetic state was the same in *Fmr1* vs WT mice?

Response: We meticulously followed the same procedure for both genotypes and gave doses of anesthesia proportional to the body weight of the mice, which was not significantly different between both groups. The desired level of anesthesia was determined by the absence of any reflexes (including eye blink, whisker movement). Additional injections of anesthesia were given when the animals started to show any reflexes. In addition to reflexes, we also monitored other physical indicators of anesthesia state such as the state of relaxation, and breathing regularity. We cannot completely exclude differences in the metabolism of the anesthetics between the genotypes, but the average interinjection interval was not different.

6. The intrinsic excitability experiments (Fig. 3D-H) were presumably performed with network activity intact (i.e., no synaptic blockers). Thus higher spontaneous network activity could have contributed to the increased F-I curves in *Fmr1* (Fig. 3E).

Response:

Although spontaneous AP firing is significantly increased in the *Fmr1*^{-y} neuron population, the values are still low (in the range of 0.2-0.5 Hz). While we cannot fully exclude the rare appearance of spontaneous APs during our i/v curve measures, it is likely not the case.

7. Figure 6 does not seem useful. Box 1 lists the only two possible sources of variability in neurons. Box 2 is a fully connected graph suggesting that all noise sources drive each other. Box 3 lists both increased variability and reduced reliability as separate consequences. But what is reduced reliability except increased variability? Most of the legend is a lengthy claim about the potential implications of the model for clinical and preclinical research, which does not seem appropriate for a figure legend.

Response:

We initially included this figure in the hope to provide a simplified synthesis of our complex findings, which may not be readily accessible to non-specialists, and to illustrate their potential link to trial-by-trial and inter-individual variability in autism. We still believe that this could be a useful addition, but would be happy to move the figure to the supplement (new Fig. S1), or remove it, if the reviewer wishes so.

REVIEWER COMMENTS

Reviewer #1 (Remarks to the Author):

Almost all issues addressed. I have 2 comments. The first relating to one remaining concern.

Previous major Comment:

1. The average differences in UP state data between WT and Fmr1 KO, as shown, do not seem large enough to account for variability in responses (i.e. driving force minimally changed).

The authors added figures 2F and 2S to response to this comment. Both indicate that the variability (or range) of EPSPs and baseline Vm are strongly correlated. These are very good data to add, and I think to address this issue empirically is very effective. The authors also emphasize that the variability in up-down state difference is higher in Fmr1 KO mice which is a good point that I missed. But I still have 2 concerns.

Variability in most plots was quantified using SD. Why do the authors switch to delta (range?) in 2F and 2S? The analysis is complicated and hard enough to read as it is without having to include new measures of variability. And wouldn't range be more subject to statistical noise? Can F and S be changed to SD bivariate plots?

Also, the authors address the question about UP state relevance by plotting delta E to delta baseline. The data are indeed consistent and suggest that UP state changes in the Fmr1 KO would impact EPSP variability. But it seems to me the more relevant plot to establish a more direct link with UP states would be delta E versus up-down-state difference range (or SD). Perhaps replace S with such a plot (S and F seem so close in what they are plotting so losing the current plot may not make much difference?).

New comment, minor

In discussion or results, please cite Rotschafer et al., 2013 as a study that first (?) observed greater AP latency variability in response to sensory stimuli in cortex of Fmr1 KO mice. I think this is the case.

Reviewer #3 (Remarks to the Author):

The authors have resolved most of the prior issues, and this revised paper should be a very useful contribution to the *Fmr1* field (and broader autism field). But one problematic interpretation issue remains that needs to be resolved (Point 1). Regarding the tape removal assay, I have no objection to their adding these new results, but additional mice are probably needed because this seems seriously underpowered (Point 2). I also think that they need to acknowledge the interpretational issues with anesthesia (Point 3).

1. The authors have backed away from most of their claims about inter-individual variability in their cellular measures. However, they still report this data (Table S3) and state that some of their measures show more variability from mouse to mouse. But there is a serious problem here: the data are not appropriate to make claims about mouse-to-mouse variability, because there were very few cells per mouse (usually only 1, according to the rebuttal letter--I couldn't find the exact N per mouse stated in the paper). In this regime, the calculated mouse-to-mouse variability will largely reflect cell-to-cell variability, and is not a meaningful estimate of true mouse-to-mouse variability. Thus, one simply cannot tell from this dataset whether inter-individual variability is elevated in *Fmr1*. Unfortunately, I think the mouse-to-mouse variability analysis, and claims about it, should be removed. This is not a limitation for the behavior, only for the cell physiology.

2. The authors ask whether they should include the new tape removal assay. While it has some interpretational unknowns (is variance due to sensory function? motor function? difference in internal motivation state?), I do think it would help support the inter-individual variability argument. But the N needs to be substantially increased to appropriately power this test. A quick power analysis (using *G_Power*, for a F-test of equal variance between two samples) shows that to statistically detect a 3:1 variance ratio between groups (as in Reviewer Fig. panel D) with $\alpha = 0.05$, power = 0.80, and two-tailed, would require $n=28$ mice per group. Maybe this number is not entirely accurate, but certainly more animals seem to be needed than in Reviewer Fig. panel C.

Minor

3. Anesthesia. Both Rev 2 and I brought up interpretational issues related to anesthesia. I understand that anesthesia is necessary for the experimental method. But the authors should include a sentence acknowledging that it is possible that ketamine affects WT and *Fmr1* mice differently, which could contribute to some of the phenotypes.

4. For the new acoustic startle behavioral data in Fig. 1V, individual mouse data points should be shown as well as the mean/SEM that is currently shown.

Reviewer #1 (Remarks to the Author):

Almost all issues addressed. I have 2 comments. The first relating to one remaining concern.

Previous major Comment:

1. Comment:

The average differences in UP state data between WT and Fmr1 KO, as shown, do not seem large enough to account for variability in responses (i.e. driving force minimally changed).

Response:

We agree with the reviewer that the magnitude of the changes in up–down state Vm difference is likely not the only cause of the observed variability in sensory stimulus evoked responses (although its magnitude and impact on the impedance and driving force within the dendritic tree cannot be properly addressed by somatic recordings). However, our data show a strong correlation between the membrane Vm, the membrane Vm fluctuations (including up–down state Vm difference) and various parameters of tactile stimulus evoked neuronal responses such as EPSP trial-by-trial variability. We therefore propose the up–down state Vm imbalance as one source of variability (see also next comment).

2. Comment:

The authors added figures 2F and 2S to response to this comment. Both indicate that the variability (or range) of EPSPs and baseline Vm are strongly correlated. These are very good data to add, and I think to address this issue empirically is very effective. The authors also emphasize that the variability in up-down state difference is higher in Fmr1 KO mice which is a good point that I missed. But I still have 2 concerns.

(a) Variability in most plots was quantified using SD. Why do the authors switch to delta (range?) in 2F and 2S? The analysis is complicated and hard enough to read as it is without having to include new measures of variability. And wouldn't range be more subject to statistical noise? Can F and S be changed to SD bivariate plots?

Response:

We thank the reviewer for pointing out that the use of the range (delta) can be disadvantageous. Therefore, we have changed to SD for figure 2F. We also agree with the reviewer to replace panel 2S (see next comment).

(b) Also, the authors address the question about UP state relevance by plotting delta E to delta baseline. The data are indeed consistent and suggest that UP state changes in the Fmr1 KO would impact EPSP variability. But it seems to me the more relevant plot to establish a more direct link with UP states would be delta E versus up-down-state difference range (or SD). Perhaps replace S with such a plot (S and F seem so close in what they are plotting so losing the current plot may not make much difference?).

Response:

We agree with the reviewer and have replaced figure panel 2S by plotting SD EPSP amplitude vs. up–down–state Vm difference.

New comment, minor:

In discussion or results, please cite Rotschafer et al., 2013 as a study that first (?) observed greater AP latency variability in response to sensory stimuli in cortex of Fmr1 KO mice. I think this is the case.

Response:

We thank the reviewer for this information, and have cited this reference in the discussion on page 20.

Reviewer #3 (Remarks to the Author):

The authors have resolved most of the prior issues, and this revised paper should be a very useful contribution to the Fmr1 field (and broader autism field). But one problematic interpretation issue remains that needs to be resolved (Point 1). Regarding the tape removal assay, I have no objection to their adding these new results, but additional mice are probably needed because this seems seriously underpowered (Point 2). I also think that they need to acknowledge the interpretational issues with anesthesia (Point 3).

1. Comment:

The authors have backed away from most of their claims about inter-individual variability in their cellular measures. However, they still report this data (Table S3) and state that some of their measures show more variability from mouse to mouse. But there is a serious problem here: the data are not appropriate to make claims about mouse-to-mouse variability, because there were very few cells per mouse (usually only 1, according to the rebuttal letter--I couldn't find the exact N per mouse stated in the paper). In this regime, the calculated mouse-to-mouse variability will largely reflect cell-to-cell variability, and is not a meaningful estimate of true mouse-to-mouse variability. Thus, one simply cannot tell from this dataset whether inter-individual variability is elevated in Fmr1. Unfortunately, I think the mouse-to-mouse variability analysis, and claims about it, should be removed. This is not a limitation for the behavior, only for the cell physiology.

Response:

We agree with this notion regarding the weakness of this interpretation and have therefore removed the mouse-to-mouse variability information for the cell physiology data.

2. Comment:

The authors ask whether they should include the new tape removal assay. While it has some interpretational unknowns (is variance due to sensory function? motor function? difference in internal motivation state?), I do think it would help support the inter-individual variability argument. But the N needs to be substantially increased to appropriately power this test. A quick power analysis (using G_Power, for a F-test of equal variance between two samples) shows that to statistically detect a 3:1 variance ratio between groups (as in Reviewer Fig. panel D) with $\alpha = 0.05$, power = 0.80, and two-tailed, would require $n=28$ mice per group. Maybe this number is not entirely accurate, but certainly more animals seem to be needed than in Reviewer Fig. panel C.

Response:

While our preliminary data from the tape removal test demonstrate increased variability in the Fmr1 KO mice, we agree that our data is underpowered. However, due to practical issues related to the time required to expand our breeding couples, the age of the tested mice and required n-numbers, we have decided to not include the tape removal test data in the current manuscript.

Minor comment

3. Anesthesia. Both Rev 2 and I brought up interpretational issues related to anesthesia. I understand that anesthesia is necessary for the experimental method. But the authors should include a sentence acknowledging that it is possible that ketamine affects WT and Fmr1 mice differently, which could contribute to some of the phenotypes.

Response:

We have now added a sentence acknowledging ketamine's potentially differential rate of metabolism or influence between WT and Fmr1 KO mice (page 20).

4. Comment:

For the new acoustic startle behavioral data in Fig. 1V, individual mouse data points should be shown as well as the mean/SEM that is currently shown.

Response:

We thank you for this suggestion, and we have now amended the graph to show individual data points (Fig. 1V).

REVIEWERS' COMMENTS

Reviewer #1 (Remarks to the Author):

All my concerns are addressed.

Reviewer #3 (Remarks to the Author):

The authors have addressed all my comments. The results are now clear and I congratulate the authors on a very nice study.